# The DREAM complex functions as conserved master regulator of somatic DNA-repair capacities

Arturo Bujarrabal-Dueso [1,2], Georg Sendtner[1,2], David H. Meyer [1,2], Georgia Chatzinikolaou[3], Kalliopi Stratigi [3], George A. Garinis [3] & Björn Schumacher [1,2] ✉

The DNA-repair capacity in somatic cells is limited compared with that in germ cells. It has remained unknown whether not only lesion-type-specific, but overall repair capacities could be improved. Here we show that the DREAM repressor complex curbs the DNA-repair capacities in somatic tissues of *Caenorhabditis elegans*. Mutations in the DREAM complex induce germline-like expression patterns of multiple mechanisms of DNA repair in the soma. Consequently, DREAM mutants confer resistance to a wide range of DNA-damage types during development and aging. Similarly, inhibition of the DREAM complex in human cells boosts DNA-repair gene expression and resistance to distinct DNA-damage types. DREAM inhibition leads to decreased DNA damage and prevents photoreceptor loss in progeroid *Ercc1⁻/⁻* mice. We show that the DREAM complex transcriptionally represses essentially all DNA-repair systems and thus operates as a highly conserved master regulator of the somatic limitation of DNA-repair capacities.

Genomes are constantly exposed to exogenous and endogenous genotoxic insults. DNA-repair proficiency depends on cell type and cell cycle stage, and is particularly different in germ cells than in somatic cells. In germ cells, DNA repair is highly efficient to maintain genome integrity throughout generations; in somatic cells, DNA repair maintains genome integrity early in life, but operates inefficiently during later, post-reproductive stages[1]. Although germlines have mutation rates that are orders of magnitude lower than those in somatic tissues[2–4], the high mutation rates in the soma increase in an age-dependent manner across species[5]. In *C. elegans*, somatic cells mostly terminally differentiate during embryogenesis and are entirely post-mitotic in the adult, whereas germ cells retain mitotic and meiotic activity. Germ cells survey their genome for helix-distorting lesions by global-genome nucleotide excision repair (GG-NER) and accurately repair DNA double-strand breaks (DSBs) through homologous recombination repair (HRR)[6–10].

In somatic cells, GG-NER[11] and HRR[8,12–14] are dispensable, as they instead use error-prone non-homologous end joining (NHEJ), and only actively expressed genes are surveilled by transcription-coupled NER (TC-NER)[7,9]. The resistance to DNA-damage-driven developmental growth impairment and functional deterioration during aging is thus limited by the restriction of somatic DNA-repair capacities. Also in mammals, the engagement of accurate DNA-repair systems depends on the cell cycle and differentiation state[15]. Not only HRR, but also additional repair pathways, are enhanced during replication, like single-strand annealing, microhomology end joining and long-patch base-excision repair (BER)[16–18]. By contrast, cell types that are either quiescent, terminally differentiated, or senescent have limited DNA-repair capacities.

Cellular quiescence and differentiation are controlled by the DREAM complex, formed by the Dp/Retinoblastoma(Rb)-like/E2F and the MuvB subcomplexes[19]. In *C. elegans*, the DREAM complex

[1]Institute for Genome Stability in Aging and Disease, Medical Faculty, University and University Hospital of Cologne, Cologne, Germany. [2]Cologne Excellence Cluster for Cellular Stress Responses in Aging-Associated Diseases (CECAD), Center for Molecular Medicine Cologne (CMMC), University of Cologne, Cologne, Germany. [3]Institute of Molecular Biology and Biotechnology, Foundation for Research and Technology-Hellas, Department of Biology, University of Crete, Heraklion, Crete, Greece. ✉e-mail: bjoern.schumacher@uni-koeln.de

comprises subunits encoded by genes that were first discovered as synthetic multivulva (synMuv) class B genes, owing to their role in cellular differentiation in combination with mutant proteins encoded by other synMuv gene classes[20–22]. Single mutations in these genes were sufficient to promote germline-like characteristics in the soma, including misexpression of germline genes[23–26]. In humans, in addition to the highly conserved repressor function in quiescence, the components of the DREAM complex can associate with other proteins and instead function as a transcription activator during the cell cycle[27–29]. The specific assembly of the repressive DREAM complex in G0 is regulated by the DYRK1A protein kinase, which, through phosphorylating LIN52, can bind to p130 and form the complex[30].

Here, we report that the DREAM complex represses a wide range of DNA-repair genes in the soma of *C. elegans*. Mutations in genes encoding DREAM components trigger DNA-damage resistance in somatic tissues against a broad range of genotoxic insults, including ultraviolet (UV) lesions, alkylations, interstrand crosslinks (ICLs) and DSBs. DREAM mutants showed accelerated lesion removal and suppressed the sensitivity to DNA damage of animals deficient in DNA-repair genes. In human cells, the DYRK1A inhibitors harmine and INDY[31,32] triggered the induction of DREAM-targeted DNA-repair genes and conferred resistance to distinct DNA-damage types. In vivo, harmine treatment reduced DNA damage and apoptosis in the retinas of *Ercc1⁻/⁻* progeroid mice. We thus establish that pharmacologically targeting the DREAM complex could be applied to augment genome stability. We propose that the DREAM complex represses the expression of various DNA-repair mechanisms and thus limits DNA-repair capacities. Therefore, inhibition of DREAM could overcome the consequence of dysfunction in single DNA-repair systems and DNA-damage-driven aging.

## Results

### DDR gene promoters carry the CDE-CHR DREAM-binding motif

To investigate the mechanisms underlying transcriptional regulation of DDR genes in *C. elegans*, we assessed whether specific transcription-factor-binding sites might be overrepresented in DDR gene promoters. An unbiased DNA-motif enrichment analysis of the 211 DDR genes (Supplementary Table 1) revealed a significant enrichment of the DPL-1-, EFL-1- and LIN-15B-binding motifs (Fig. 1a). DPL-1 and EFL-1 form the E2F–DP heterodimer, which directly contacts promoters by binding to the cycle-dependent element (CDE). E2F–DP is linked through the pocket protein LIN-35 to the MuvB subcomplex, which binds the cell cycle genes homology region (CHR) in promoters to then form the DREAM transcriptional repressor complex[33–35]. We identified the CDE-CHR motif in the promoters of 125 of the 211 DDR genes (Fig. 1b), suggesting that the DREAM complex is a regulator of DDR genes.

### DREAM-complex mutants confer DNA-damage resistance

We next determined whether mutations in DREAM components influence DNA-damage sensitivity. We tested UV-induced DNA lesions because they impact developmental growth and the longevity of the animals[9]. UV-B irradiation induces formation of cyclobutane pyrimidine dimers (CPDs) and pyrimidine (6–4) pyrimidone photoproducts (6–4PPs) that are repaired by NER. Except for the two primordial germ cells, all 558 cells of the L1 larvae are somatic cells, of which 90% are terminally differentiated[36]. We exposed synchronized DREAM-mutant L1 larvae to UV and scored developmental growth 48 hours (h) later, counting the developmental stages for all the worms, from L1 to the consecutive L2, L3 and L4 larval and adult stages. Surprisingly, worms with loss-of-function mutations in *lin-52*, *dpl-1*, *efl-1*, *lin-53* or *lin-35*, which encode DREAM-complex components, showed a significant improvement in somatic development compared with wild-type (WT) worms following UV exposure (Fig. 1c and Extended Data Fig. 1a). Even though *lin-35* mutants showed a developmental delay in the absence of UV[26], they proceeded through development more rapidly than did WT worms at high UV doses (Fig. 1c).

Double-mutant worms, with mutations in two genes (*lin-52*; *dpl-1* and *lin-52*; *efl-1*), showed improved developmental growth, similar to that in *lin-52* mutants, indicating that their encoded DREAM-complex subunits conferred UV resistance (Fig. 1d and Extended Data Fig. 1b). By contrast, worms with mutations in the synMuv B class of genes or a mutant component of the chromatin-remodeling NuRD complex that is not part of the DREAM complex did not have improved developmental growth upon UV exposure (Extended Data Fig. 1c). These data suggest that the specific function of LIN-52, DPL-1, EFL-1, LIN-53 and LIN-35 as subunits of the DREAM complex determine the animals' ability to overcome DNA-damage-induced developmental delay.

To evaluate whether mutations in the DREAM-complex genes could affect DNA-damage-driven organismal aging, we UV-treated the DREAM-complex-mutant worms on day 1 of adulthood and assessed their lifespans (Fig. 1e). In humans, mutations in DNA-repair genes are sufficient to accelerate aging and lead to premature death[37], but *C. elegans* worms cultured under laboratory conditions require exogenous DNA damage to shorten their lifespans[9,38,39]. Worms with mutations in *lin-52*, *dpl-1*, *efl-1* or *lin-35* significantly outlived WT worms upon DNA damage, despite the fact that some were short-lived without irradiation.

The reduced lifespans of *dpl-1*, *elf-1* and *lin-35* mutant animals under unperturbed conditions might be connected to their previously described roles in contributing to developmental processes, whereas *lin-52(n771)* might be a hypomorphic mutation that does not affect associations among the complex components[40]. In the absence of genotoxins, worms with the *lin-52(n771)* genotype showed only a slight reduction in egg-laying capacity and a mild sensitivity to starvation

**Fig. 1 | Mutations in genes encoding components of the DREAM complex confer resistance to UV-induced DNA damage during development and adulthood. a**, Sequences of the motifs found upon analysis of the promoters of the DDR genes using HOMER. BG, background; FC, fold change; E2F, E2F transcription factor; Zf, zinc finger domain. **b**, Result of the motif search for the CDE-CHR DREAM complex motif in the promoters of the DDR genes using HOMER. **c,d**, UV-irradiation assay during somatic development of WT, *lin-52(n771)*, *lin-35(n745)*, *dpl-1(n2994)*, *efl-1(se1)* (**c**) and WT, *lin-52(n771)*, *dpl-1(n2994)* and *lin-52(n771)*; *dpl-1(n2994)* mutant worms (**d**). *Y* axis shows the percentage of the different larval stages, and *x* axis the UV dose applied in mJ/cm². Representative graph showing *n* = 3 biological replicates from 1 of at least 3 independent experiments. Data are shown as mean ± s.d. of each larval stage (L1–L4-A). For statistical analysis, a two-tailed *t*-test between the fraction of each larval stage of a mutant compared with WT in the same treatment condition was used, except for *lin-52(n771)*; *dpl-1(n2994)*, which was compared with *lin-52(n771)*. **e**, Lifespan assay upon exposure of WT and DREAM-complex-mutant worms to UV-B (0 and 400 mJ/cm²). A log-rank test was performed to compare the lifespan of the DREAM mutants and WT worms in the same conditions. Top graphs, *n* = 122 (without UV), 150 (with UV) for WT; 121, 152 for *lin-52*; and 128, 154 for *dpl-1* mutants. Bottom graphs, *n* = 138, 218 for WT; 162, 215 for *efl-1* and 144, 201 for *lin-35* mutants. Bar graphs show the percentage by which mean lifespan decreased for each strain irradiated with UV-B compared with the mock-treated worms of the same strain. **f**, UV-irradiation assay for germline development of WT, *xpc-1(tm3886)*, *lin-52(n771)* and the double mutant, *lin-52(n771)*; *xpc-1(tm3886)*. Representative graphs of *n* = 3 biological replicates, from 1 of 3 independent experiments. The mean ± s.d. of eggs laid or the percentage hatched is shown. Two-tailed *t*-tests were performed to compare the number of eggs laid and hatched between the different strains within the same condition. For **a** and **b**, the *P* value over the background was calculated with a hypergeometric test, and the *q* value shows the Benjamini–Hochberg-adjusted *P* values. For **c** and **d**, *P* > 0.05, not shown; *\**P* < 0.05, **\*\**P* < 0.01, ***\*\*\**P* < 0.001, \*\*\*\* *P* < 0.0001. Remaining comparisons and detailed *P* values are provided in Supplementary Table 13, including results from Fisher's exact test.

(Extended Data Fig. 2a,b). This suggests that a partial loss of function of DREAM's function enhances the resistance to DNA damage without affecting other physiological processes.

The lifespan reduction under unperturbed conditions is consistent with the reported lifespan shortening of *lin-9*, *lin-35* and *lin-37* mutant animals[41]. Another report has found that *lin-52*, *lin-37* or *dpl-1* mutant

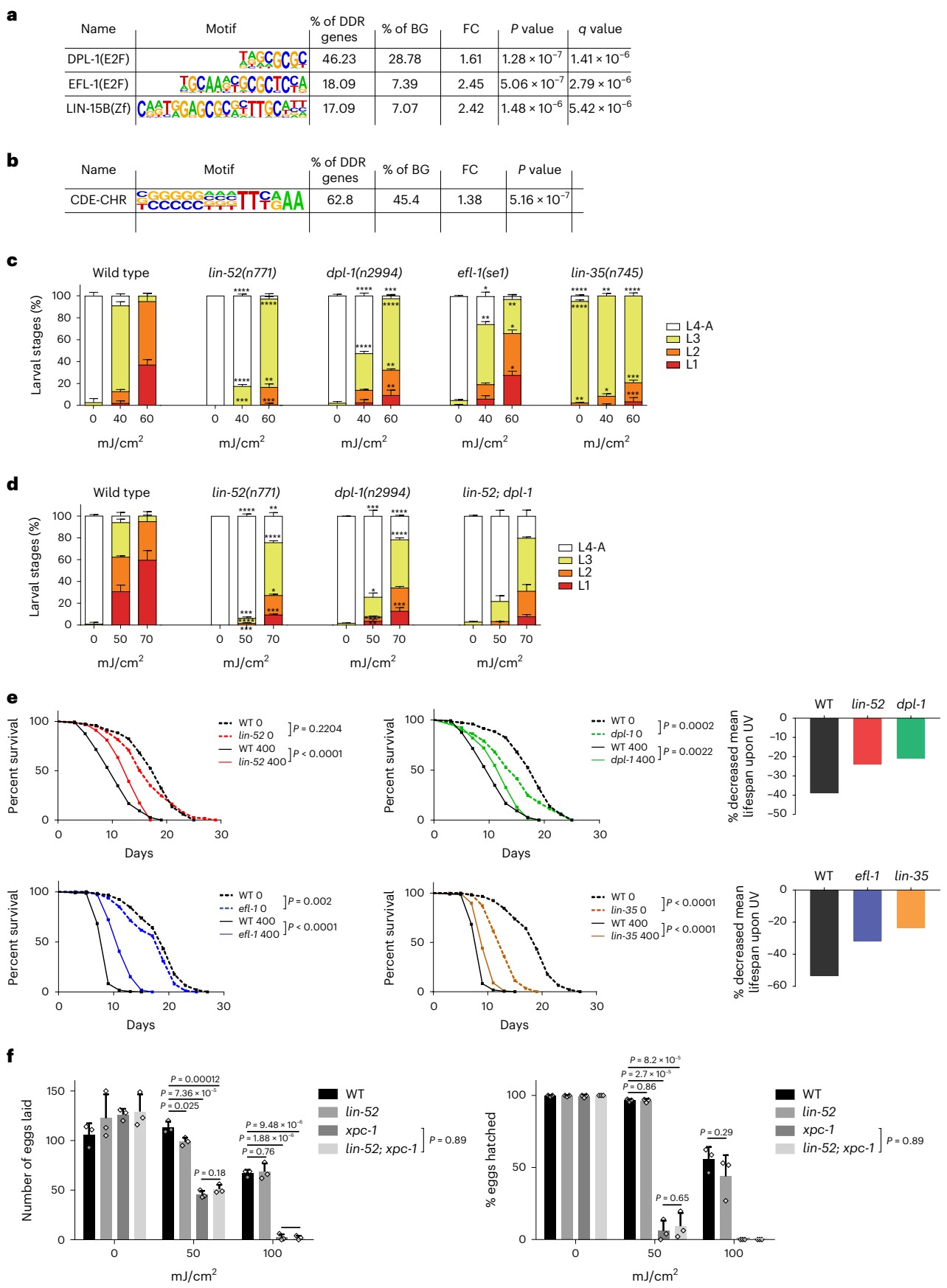

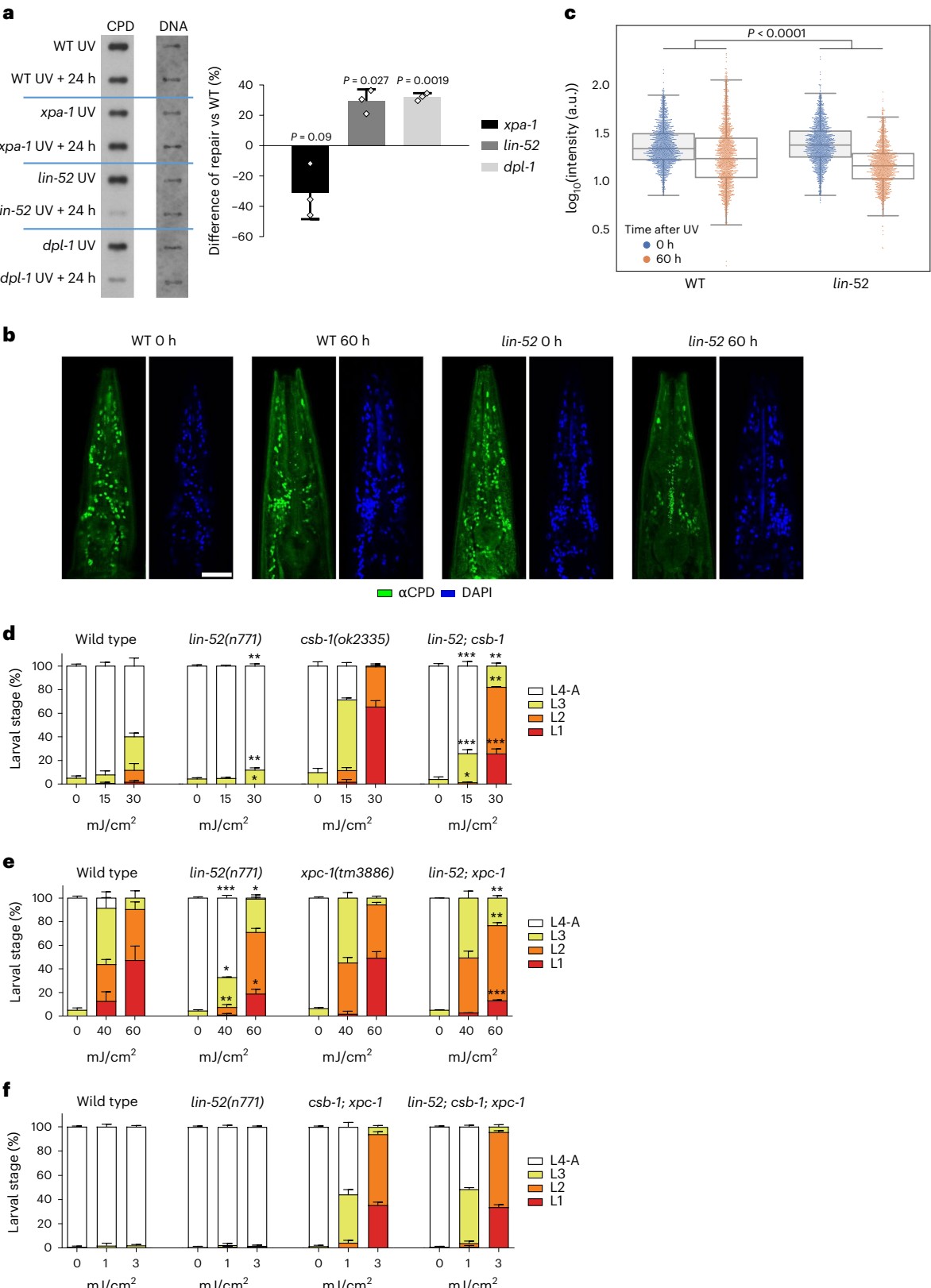

worms treated with 5-fluoro-2′-deoxyuridine (FUdR)[42], which is genotoxic, had extended lifespans, further supporting our findings that DREAM mutants alleviate DNA-damage-induced lifespan shortening.

Indicative of healthspan extension, *lin-52* mutants animals retained more motility than did WT animals (Extended Data Fig. 3). Owing to the phenotypic specificity with mild adverse effects under unperturbed conditions and the strong UV-resistance phenotype, we

decided to focus mainly on LIN-52 mutants to further investigate the role of the DREAM complex in regulating genome stability.

Considering that DREAM represses gene expression in non-dividing cells[19], we hypothesized that the UV sensitivity in the germline would be unaffected. *lin-52* mutant animals laid a comparable amount of eggs to WT worms upon UV exposure, with similar hatching rates, and the *lin-52* mutation did not alleviate the germline

**Fig. 2 | DREAM-complex mutants show enhanced repair of UV-induced DNA lesions and alleviate the UV sensitivity of *csb-1* and *xpc-1* mutant animals.** **a**, DNA-repair capacity assay in WT, *xpa-1(ok698)*, *lin-52(n771)* and *dpl-1(n2994)* L1 worms. A representative slot blot of three independent experiments is shown. Samples labelled as 'UV' were collected right after UV irradiation; samples 'UV + 24 h' were collected 24 h after UV irradiation. Graphs show the mean ± s.d. of the improved or decreased repair of the mutants compared with that of WT worms. *n* = 3 biological replicates. A two-tailed *t*-test was used to compare the mutants' repair with that of WT worms. **b**, Representative images of a focal plane of the anterior region of adult worms irradiated and stained with antibodies to CPDs and DAPI, collected right after irradiation (0 h) or after incubation for 60 h. Scale bar, 25 µm. **c**, Quantification of CPD nuclei signal intensity in the heads of adult worms irradiated and collected immediately (0 h, blue dots) or 60 h after irradiation (orange dots). The number of nuclei quantified was, at 0 h and 60 h, respectively, *n* = 1,469 and 1,479 for WT, *n* = 1.454 and 1.321 for *lin-52(n771)*, from 5–7 heads per condition. The *y* axis shows the $\log_{10}$-transformed intensity values of CPDs. Box midlines show the median, box limits show the top and

bottom quartiles and whiskers extend to 1.5 × interquartile range (IQR). Two-way analysis of variance (ANOVA) between the strain (WT and *lin-52*) and the time component is shown. a.u., arbitrary units. **d**–**f**, UV-irradiation assay during somatic development of WT, *lin-52(n771)*, *csb-1(ok2335)* and *lin-52(n771)*; *csb-1(ok2335)* (**d**); WT, *lin-52(n771)*, *xpc-1(tm3886)* and *lin-52(n771)*; *xpc-1(tm3886)* (**e**); and WT, *lin-52(n771)*, *csb-1(ok2335)*; *xpc-1(tm3886)* and *lin-52(n771)*; *csb-1(ok2335)*; *xpc-1(tm3886)* (**f**). *Y* axis shows the percentage of the different larval stages, and *x* axis the UV dose applied in mJ/cm². Graphs are representative of *n* = 3 biological replicates from 1 of 3 independent experiments. Data are shown as mean ± s.d. of each larval stage. Results from two-tailed *t*-tests between the fraction of each larval stage of *lin-52*-mutated worms compared with WT, and each larval stage of *lin-52*-mutated NER-deficient worms compared with the NER-deficient control, are shown for the same treatment conditions. *P* > 0.05, not shown. \**P* < 0.05, \*\**P* < 0.01, \*\*\**P* < 0.001, \*\*\*\**P* < 0.0001. For **d**–**f**, detailed *P* values and comparisons against WT worms are in Supplementary Table 13, including results from Fisher's exact test to analyze the overall distribution of the larval stages.

hypersensitivity of worms with mutated *xpc-1* (Fig. 1f). Therefore, DREAM mutations specifically augment DNA-damage resistance of the somatic tissues.

## DREAM-complex mutants improve DNA repair

To test whether DREAM mutants enhance DNA repair, we measured the removal of the main UV-induced DNA lesion type, CPDs. Twenty-four hours following UV treatment of L1 larvae, we quantified CPDs using an anti-CPD antibody. The tested DREAM mutants showed significantly improved CPD repair compared with WT animals (Fig. 2a). We confirmed the improved repair by using an anti-6-4PP antibody, which also revealed a decrease in the amount of 6-4PPs in *lin-52* mutant animals (Extended Data Fig. 4a).

To exclude that enhanced lesion removal might be a consequence of damage dilution due to DNA replication, we performed an EdU-incorporation assay in L1 worms. Both *lin-52* mutant and WT worms showed comparable DNA-replication events within 24 h of UV exposure, further confirming that the observed decrease in CPD and 6-4PP was due to repair (Extended Data Fig. 4b).

We next assessed whether the DREAM complex also regulates the DNA-repair capacity in adult animals. CPD quantification in the nuclei-dense heads of UV-treated worms at day 1 of adulthood revealed an improved removal of CPD lesions in *lin-52* mutants compared with that in WT worms, indicating an augmented repair capacity (Fig. 2b,c). EdU incorporation in intestinal cells, which are among the most prone to hyperproliferate[43] (Extended Data Fig. 4c) showed no replication events that could alter the response to DNA damage.

To determine whether the effect of *lin-52* on DNA repair was specific to the somatic cells, we performed the same immunofluorescence DNA-repair analysis on germlines upon UV exposure. Both WT and *lin-52* mutant worms had a similar, highly efficient repair capacity

(Extended Data Fig. 5a). Therefore, the improved capacity to repair CPDs is specific to the somatic cells of *lin-52* mutants.

NER is initiated either by the TC-NER protein CSB-1 in the transcribed strand, which is particularly relevant in somatic tissues, or by the GG-NER protein XPC-1, which recognizes damage throughout the whole genome and is crucial in the germline[7]. Both branches then recruit XPA-1 to assemble the NER core machinery. We evaluated whether mutations in the DREAM complex required these NER branches in order to confer resistance to UV-induced DNA damage. The *lin-52* mutation alleviated the UV sensitivity of *csb-1*, *csa-1* and *xpc-1* mutants (Fig. 2d,e and Extended Data Fig. 5b) but had no effect on completely NER-deficient *xpa-1* mutants or *csb-1*; *xpc-1* double mutants (Fig. 2f and Extended Data Fig. 5c). Therefore, a mutation in the DREAM complex improved both GG-NER and TC-NER and could thus partially compensate for defects in either one of the NER-initiating systems, whereas the enhanced UV resistance depends on the presence of the NER machinery.

## The DREAM complex represses multiple DNA-repair pathways

To address whether the DREAM complex could curb the repair capacity of somatic cells by directly repressing DNA-repair genes, we performed RNA sequencing (RNA-seq) of *lin-52(n771)* and WT L1 larvae (Fig. 3). The majority of differentially expressed genes were upregulated in *lin-52* mutants, and these genes included a range of genes involved in DNA-repair mechanisms (Fig. 3a, Table 1 and Supplementary Table 2). Gene Ontology (GO) analysis of the significantly upregulated genes (adjusted *P* < 0.05) in *lin-52* mutants compared with WT were strongly enriched for DDR-related terms (Fig. 3b and Supplementary Table 3). Similarly, proteomic analysis of *lin-52* mutant and WT worms revealed that multiple DNA-repair proteins were upregulated (Fig. 3c) and multiple GO terms related to DNA repair were also

**Fig. 3 | The DREAM complex directly represses multiple DNA-damage-response genes that are normally enriched in the germline. a**,**c**, Differentially expressed genes (**a**) or proteins (**c**) in *lin-52(n771)* mutants (adjusted *P* < 0.05). DDR genes (or their products) are shown in orange, and the significantly changed DDR genes in common between **a** and **c** are labeled. **b**,**d**, GO enrichment analysis for the upregulated genes (**b**) or proteins (**d**) in *lin-52(n771)* mutants compared with WT (adjusted *P* < 0.05, two-sided Fisher's exact test with FDR). Highly overlapping terms were removed for simplicity. Terms related to DNA-damage responses are shown in red. The dashed lines mark an adjusted *P* value of 0.05. The full list is in Supplementary Table 3 and Supplementary Table 4. **e**, FC of genes that were significantly changed (adjusted *P* < 0.05) in both the proteome and transcriptome of *lin-52* mutants. **f**, qPCR analysis of DDR genes in *lin-52(n771)*, *dpl-1(n2994)* and *efl-1(se1)* mutants. Data are shown as mean ± s.d., *n* = 3 biological replicates. Results from two-tailed *t*-tests are in Supplementary Table 13. **g**, Overlap between the DDR genes upregulated in *lin-52(n771)* and the genes involved in the main DNA-repair pathways. Overlap between repair pathways is

not shown. Pathway genes were obtained from the GO database released on 8 October 2019. **h**, Overlap between the DDR genes that were upregulated in *lin-52* compared with WT worms and two published transcriptome datasets on *lin-35(n745)*[44,45]. **i**, GSEA of all DDR genes in the RNA-seq of *lin-52* and the genes bound by DREAM, as described in ref. [35]. Seventy-six out of the 211 DDR genes were found in both and were used for the analysis. DDR genes bound by DREAM that are upregulated in *lin-52* mutants are shown in red; downregulated genes are shown in blue. NES, normalized enrichment score. **j**, Overlap between the upregulated DDR genes in *lin-52* mutants and the genes that were found to be bound by the DREAM complex in the promoter area (41 in promoter area, 43 in total). This is a re-analysis of work in ref. [35]. **k**,**l**, Overlap between all the DDR genes in *C. elegans* (**k**) and those upregulated in *lin-52* mutants (**l**) and the genes that were enriched in the germline in ref. [46]. **m**, GSEA of the RNA-seq of *lin-52(n771)* with the genes enriched in the germline[46]. Results from two-sided Fisher's exact test are shown for overlap analyses. All GSEA statistics were done as described in ref. [77].

enriched (Fig. 3d and Supplementary Table 4) in the mutants. The significantly regulated genes showed very similar regulation in both the transcriptome and proteome, with multiple DDR genes induced in both (Fig. 3e).

We also determined by quantitative PCR (qPCR) that DDR genes were upregulated in other DREAM-complex mutants (Fig. 3f). A total of 53 DDR genes were significantly upregulated (adjusted $P < 0.05$) in *lin-52* mutants compared with WT worms (Table 1). Among these were genes involved in NER, ICL repair, BER, HRR, mismatch repair and NHEJ,

suggesting that the DREAM complex represses components of all main DNA-repair pathways (Fig. 3g).

We next searched for DNA-repair genes in other DREAM complex transcriptomic data. The induced DNA-repair genes in *lin-52* mutants showed a remarkably consistent induction in two transcriptome datasets of *lin-35(n745)* mutants[44,45]. Out of the 53 DDR genes induced in *lin-52* mutants, 35 were also found in *lin-35* L1s, and 44 in *lin-35* L3 worms (Fig. 3h and Table 1). This overlap in two independent studies further substantiates that the DREAM complex regulates an extensive amount of DDR genes.

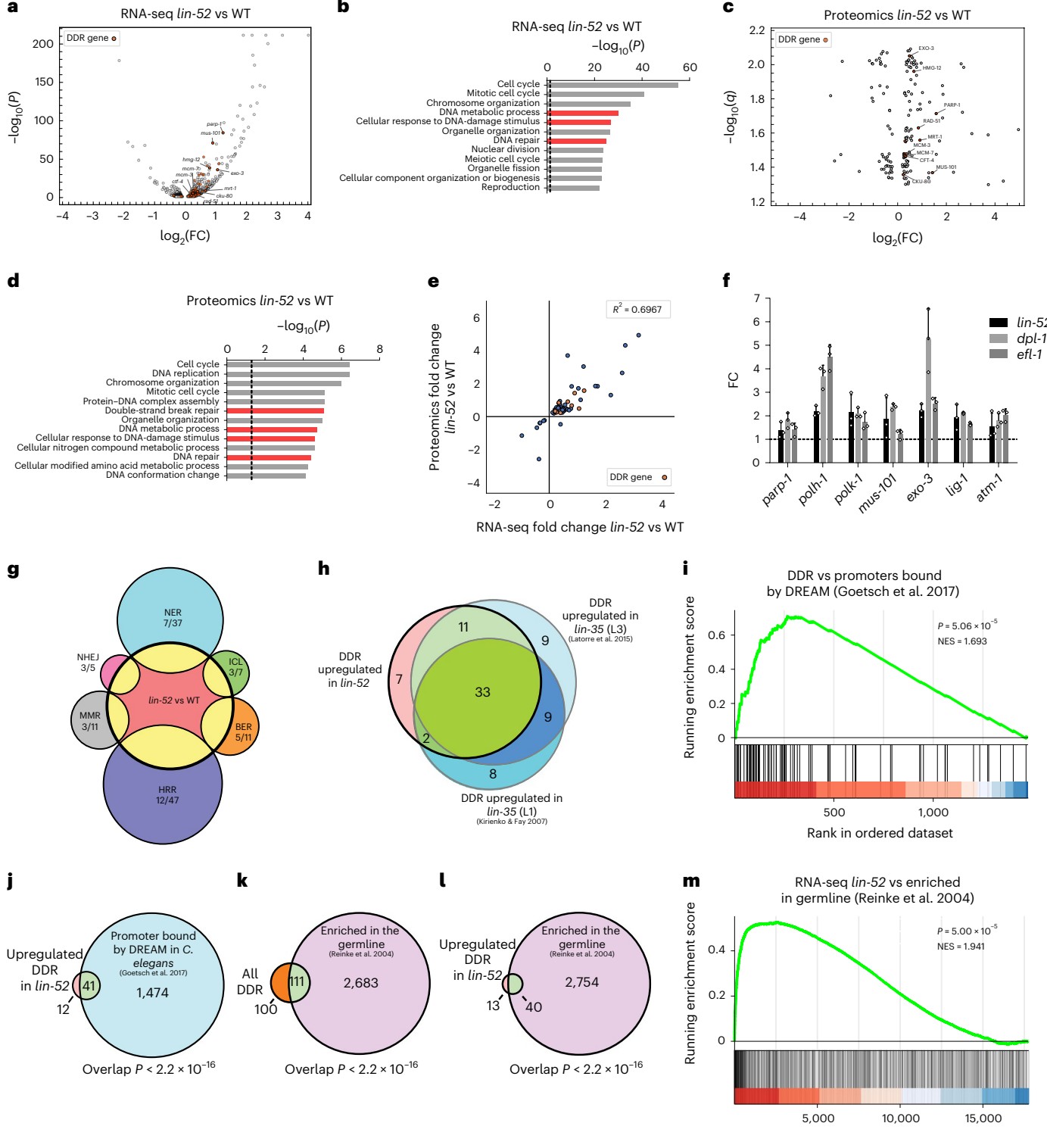

**Table 1 | The DREAM complex binds and represses DDR gene expression**

| Gene | FC lin-52 vs WT | Adjusted P | FC lin-35 vs WT (L1)[45] | FC lin-35 vs WT (L3)[44] | Bound by DREAM[35] |
|---|---|---|---|---|---|
| atm-1 | 1.53 | $1.91 \times 10^{-53}$ | NA | 1.52 | ✓ |
| baf-1 | 1.15 | $2.96 \times 10^{-3}$ | NA | 1.35 | ✓ |
| brc-1 | 1.34 | $2.39 \times 10^{-6}$ | 1.95 | 2.57 | ✓ |
| brd-1 | 1.23 | $9.11 \times 10^{-3}$ | 1.78 | 2.39 | ✓ |
| chk-1 | 1.18 | $1.39 \times 10^{-2}$ | 2.18 | 1.73 | ✓ |
| cku-80 | 1.32 | $1.63 \times 10^{-8}$ | 2.43 | 2.31 | ✓ |
| clsp-1 | 1.21 | $6.22 \times 10^{-5}$ | 2.27 | 2.59 | ✓ |
| crn-1 | 1.30 | $7.34 \times 10^{-11}$ | 1.57 | 1.96 | ✓ |
| csa-1 | 1.38 | $1.15 \times 10^{-4}$ | 4.37 | 4.87 | ✓ |
| ctf-4 | 1.16 | $2.23 \times 10^{-3}$ | 2.07 | 1.70 | ✓ |
| dog-1 | 1.27 | $1.74 \times 10^{-5}$ | 1.76 | 2.23 | ✓ |
| exo-1 | 1.27 | $4.24 \times 10^{-5}$ | 2.64 | 2.43 | ✓ |
| exo-3 | 2.08 | $4.70 \times 10^{-37}$ | 2.99 | 2.07 | ✓* |
| F10C2.4 | 1.35 | $3.99 \times 10^{-19}$ | 1.62 | 1.22 | ✓ |
| fan-1 | 1.32 | $5.34 \times 10^{-3}$ | 1.79 | 1.55 | ✓ |
| fcd-2 | 1.14 | $9.27 \times 10^{-3}$ | NA | [1.16] | ✓ |
| H21P03.2 | 1.34 | $2.97 \times 10^{-6}$ | 1.61 | 1.63 | ✓ |
| him-1 | 1.10 | $8.10 \times 10^{-3}$ | NA | [1.10] | |
| his-3 | 1.24 | $2.14 \times 10^{-23}$ | NA | NA | |
| hmg-12 | 1.74 | $1.50 \times 10^{-39}$ | 1.87 | [1.16] | |
| hpr-17 | 1.31 | $2.18 \times 10^{-3}$ | 2.17 | 1.93 | ✓ |
| hsr-9 | 1.18 | $2.03 \times 10^{-9}$ | NA | [1.15] | ✓ |
| JC8.7 | 1.24 | $1.43 \times 10^{-3}$ | NA | [1.21] | ✓ |
| lig-1 | 1.65 | $2.37 \times 10^{-42}$ | 2.93 | 2.66 | ✓ |
| M03C11.8 | 1.37 | $5.61 \times 10^{-30}$ | NA | 1.26 | ✓ |
| mcm-3 | 1.19 | $3.23 \times 10^{-6}$ | 1.80 | 2.06 | ✓ |
| mcm-4 | 1.11 | $7.38 \times 10^{-4}$ | 1.63 | 1.63 | ✓ |
| mcm-6 | 1.12 | $1.06 \times 10^{-3}$ | NA | 1.40 | ✓ |
| mcm-7 | 1.20 | $7.23 \times 10^{-10}$ | 2.53 | 2.59 | ✓ |
| mre-11 | 1.21 | $1.63 \times 10^{-4}$ | 2.15 | 1.67 | ✓ |
| mrt-1 | 1.58 | $7.63 \times 10^{-10}$ | 2.32 | 2.48 | |
| msh-6 | 1.25 | $2.43 \times 10^{-10}$ | 2.62 | 2.12 | |
| mus-101 | 1.87 | $7.35 \times 10^{-72}$ | 4.13 | 2.51 | ✓ |
| parg-1 | 1.45 | $5.98 \times 10^{-32}$ | 1.66 | 1.89 | ✓* |
| parp-1 | 2.35 | $5.46 \times 10^{-85}$ | 3.36 | 3.27 | |
| pms-2 | 1.29 | $2.58 \times 10^{-3}$ | NA | 1.35 | ✓ |
| polh-1 | 2.14 | $6.85 \times 10^{-45}$ | 3.40 | 3.57 | ✓ |
| polk-1 | 1.63 | $4.18 \times 10^{-15}$ | 2.75 | 3.18 | ✓ |
| rad-50 | 1.51 | $1.05 \times 10^{-30}$ | 2.11 | 1.35 | ✓ |
| rad-51 | 1.27 | $3.44 \times 10^{-4}$ | 1.86 | 1.97 | ✓ |
| rad-54 | 1.48 | $8.45 \times 10^{-9}$ | NA | 1.60 | |
| rnf-113 | 1.23 | $3.15 \times 10^{-4}$ | NA | 1.29 | |
| rpa-1 | 1.08 | $2.43 \times 10^{-2}$ | NA | 1.25 | ✓ |
| ruvb-1 | 1.13 | $9.05 \times 10^{-3}$ | NA | [0.92] | |
| smc-3 | 1.20 | $4.95 \times 10^{-10}$ | NA | 1.38 | ✓ |
| smc-5 | 1.42 | $4.82 \times 10^{-18}$ | 2.20 | 1.79 | ✓ |

**Table 1 (continued) | The DREAM complex binds and represses DDR gene expression**

| Gene | FC lin-52 vs WT | Adjusted P | FC lin-35 vs WT (L1)[45] | FC lin-35 vs WT (L3)[44] | Bound by DREAM[35] |
|---|---|---|---|---|---|
| smc-6 | 1.15 | $1.32 \times 10^{-3}$ | 1.56 | [1.14] | ✓ |
| sws-1 | 1.35 | $2.06 \times 10^{-4}$ | 3.93 | 3.37 | ✓ |
| tdpt-1 | 1.30 | $2.85 \times 10^{-4}$ | NA | 1.86 | ✓ |
| tim-1 | 1.27 | $2.36 \times 10^{-13}$ | 1.97 | 2.04 | ✓ |
| tipn-1 | 1.70 | $2.31 \times 10^{-22}$ | 5.34 | 6.98 | ✓ |
| trr-1 | 1.12 | $1.94 \times 10^{-7}$ | NA | [1.02] | |
| ung-1 | 1.37 | $6.70 \times 10^{-5}$ | NA | 1.90 | ✓ |

Induction of the expression of DDR gene in lin-52 mutants is highly consistent in lin-35 mutants at the L1 (ref. [45]) and L3 stage[44], and promoters of these genes are bound by DREAM (re-analysis of embryonic ChIP–seq data in ref. [35]). The DDR list is based on the GO database released on 8 October 2019. P values were adjusted with the Benjamini–Hochberg FDR adjustment, calculated as described in ref. [76]. NA, not applicable, owing to the gene information not being available or in the dataset. FC values in brackets were non-significant. ✓, bound by at least 6 DREAM components in the promoter; ✓*, bound by at least 6 DREAM components in intronic or intergenic areas.

To address whether the induced DDR genes might be directly repressed by the DREAM complex, we analyzed a published chromatin immunoprecipitation and sequencing (ChIP–seq) dataset on the DREAM complex in late embryos[35]. Gene Set Enrichment Analysis (GSEA) of all the DDR genes found in the lin-52 RNA-seq data revealed a significant enrichment of DDR genes bound by DREAM (Fig. 3i and Supplementary Table 5), with 80 of the 211 DDR genes being bound by DREAM (76 in the promoter, Supplementary Table 6). Forty-three out of the 53 DDR genes that were significantly upregulated in lin-52 mutants were bound by the DREAM complex (41 in the promoter area, 2 intergenic or intronic) by at least 6 out of 7 DREAM components tested[35] (Fig. 3j and Table 1). We further confirmed the direct binding of DREAM to DDR genes by analyzing a ModENCODE ChIP–seq dataset for DREAM in L3-stage worms[44]. GSEA of all the DDR genes in the lin-52 RNA-seq data showed an enrichment of upregulated DDR genes in the list of genes bound by DREAM (Extended Data Fig. 6a and Supplementary Table 5). Sixty-three DDR genes were bound by at least one component of the DREAM complex (Supplementary Table 7). Furthermore, 33 of the 53 DDR-related genes that were significantly upregulated in lin-52 mutants were bound by at least one member of the DREAM complex, and 16 were bound by all of them (Extended Data Fig. 6b and Supplementary Table 8).

These analyses reveal that the DREAM complex directly binds and represses multiple genes involved in the DDR. The consistency of the induction of DDR genes in DREAM mutants, the direct binding of DREAM components to DDR gene promoters across different studies and the DDR protein upregulation observed indicate that DREAM constitutively represses DDR genes in somatic cells.

**Germline-like expression signature of DNA-repair genes**

Considering that the DREAM complex represses gene expression in somatic cells, we wondered whether the upregulation of DNA-repair genes in DREAM-complex mutants would resemble expression patterns of the germline[46]. We found that 111 of the 211 DDR genes were enriched in the germline (Fig. 3k and Supplementary Table 9), including 40 of the 53 DDR genes that were significantly upregulated in lin-52 mutants (Fig. 3l and Supplementary Table 9). Germline-specific genes were strongly enriched among the upregulated genes in lin-52 mutants (Fig. 3m and Supplementary Table 5); 271 out of 671 of them were germline-enriched genes, whereas only 26 out of 464 genes downregulated in lin-52 mutants were germline-enriched (Extended Data Fig. 6c and Supplementary Table 2).

Finally, we analyzed available data from microarrays of lin-54-mutant embryos and germline tissue[47]. Consistently, in lin-54 mutant

embryos, DDR genes were upregulated (Extended Data Fig. 6d and Supplementary Tables 5 and 10), and no DDR genes were downregulated. The GO analysis of the upregulated genes showed a highly significant over-representation of 'Cellular response to DNA damage' (Supplementary Table 11). By contrast, the germline of *lin-54* mutants did not present any enrichment of DDR genes compared with WT germlines (Extended Data Fig. 6e and Supplementary Table 5). These data indicate that DNA-repair genes whose expression in WT animals is restricted to the germline are particularly upregulated in somatic tissues of DREAM mutants.

In conclusion, our analysis of transcriptomic and ChIP–seq data indicates that the DREAM complex represses genes in the soma that are usually expressed in the germline. These genes are highly enriched in DDR genes, and thus a mutation in *lin-52* leads to a germline-like upregulation of DDR genes in the soma.

### DREAM mutants confer resistance to various types of DNA damage

On the basis of the range of DNA-repair pathways induced in the soma of *lin-52* mutants, we hypothesized that DREAM mutants would show resistance to a wide variety of DNA-damaging insults that require different repair machineries. Somatic cells in the very early embryo are highly replicative and, upon ionizing radiation (IR), repair the DSBs through HRR, which is initiated by the BRC-1–BRD-1 complex[13,14,48]. The hatching of larvae from IR-treated eggs was evaluated for WT; the *lin-52(n771)* mutant; HRR-deficient *brc-1(tm1145); brd-1(dw1)* and *lin-52; brc-1; brd-1* mutants; NHEJ-deficient *cku-70(tm1524)* and *lin-52; cku-70* mutants; and the *lin-52; brc-1; brd-1; cku-70* mutant, deficient for HRR and NHEJ (Fig. 4a). The proportion of egg hatching upon IR in the *lin-52(n771)* mutant was significantly higher than in WT worms. As early embryos predominantly employ HRR instead of NHEJ to repair DSBs[13,14,48], *brc-1; brd-1* HRR-deficient mutant eggs were highly IR sensitive, whereas NHEJ-deficient *cku-70* mutants had similar IR sensitivity to that of WT worms. The IR resistance conferred by mutant *lin-52* in early embryos suggests that the repression of DNA-repair genes is a property of a somatic function of DREAM and is not just associated with cellular quiescence or terminal differentiation.

Mutated *lin-52* rescued the IR sensitivity of *brc-1; brd-1* double mutants to levels similar to that of WT animals at high doses, suggesting that, in the absence of HRR, a mutation in *lin-52* leads to the induction of alternative DSB-repair pathways. *lin-52; cku-70* mutants also had a higher survival of embryos than did *cku-70* worms. However, when *lin-52* mutants were deficient in both NHEJ and HRR, we could no longer observe an improvement compared with *brc-1; brd-1* mutants. Therefore, NHEJ is required for the rescue of the embryonic survival of *brc-1; brd-1* by *lin-52*. These results suggest that *lin-52* mutants have highly efficient HRR and NHEJ pathways that can compensate for the absence of either of these pathways (Fig. 4a).

Similar to *lin-52* mutants, the increased survival upon IR treatment was also observed in *efl-1(se1)* mutant embryos (Extended Data Fig. 7a).

However, this could not be tested in other DREAM mutant strains, such as *dpl-1(n2994)* and *lin-35(n745)* mutants, owing to the decreased embryonic survival in these strains without damage induction[49,50].

The somatic cells switch from HRR in the early embryo (where most cell divisions occur) to NHEJ from the late embryo and onwards[13,14]. We determined whether a deficiency in the DREAM complex would also render the worms resistant to DSBs in an NHEJ-repair-dependent fashion. WT, *lin-52(n771)* and NHEJ-deficient *cku-70(tm1524)* and *lin-52; cku-70* L1 larvae were exposed to IR, and their developmental growth was assessed 48 h later. IR-treated *lin-52* mutants had significantly improved developmental growth compared with WT worms, and this was dependent on NHEJ (Fig. 4b). We next analyzed two other strains that are sensitive to IR[51], *xpa-1(ok698)* and *polh-1(lf31)*. The *lin-52* mutation significantly rescued the IR sensitivity in both mutants (Extended Data Fig. 7b). These results suggest that mutations in *lin-52* enhance NHEJ-dependent DSB repair, resulting in augmented IR resistance in IR-sensitive strains that have the canonical NHEJ pathway intact.

We wondered whether adult DREAM mutants might also be IR resistant. Adult *C. elegans* are extraordinarily resistant to IR treatment, necessitating very high doses to induce premature death[52]. Both *lin-52(n771)* and *efl-1(se1)* worms at day 1 of adulthood that were treated with IR showed a mild but significant lifespan extension compared with WT animals (Extended Data Fig. 7c). Thus, mutations in DREAM enhance the organismal resistance to DSBs in the soma during embryonic and larval development as well as during adulthood.

Next, we evaluated alkylation damage, a complex DNA insult repaired by several mechanisms involving DNA methyltransferases, AlkB enzymes and BER[53]. L1 worms were exposed to methyl methanesulfonate (MMS), and their development was assessed 48 h later. The *lin-52* mutation led to improved development following MMS treatment and suppressed the MMS hypersensitivity of translesion synthesis DNA polymerase eta (*polh-1*) mutant animals to levels comparable to that of WT worms (Fig. 4c).

Finally, we assessed the response to cisplatin, a commonly used antitumor drug that causes intra- and ICLs[54] that are repaired by a wide range of repair pathways, including Fanconi complex proteins, HRR and NER[55]. L1 larvae were treated with cisplatin and the development of WT and *lin-52(n771)* mutant worms was evaluated 48 h later (Fig. 4d). A mutation in the DREAM complex significantly alleviated the growth retardation following cisplatin-induced DNA damage for all doses tested.

Taken together, these data show that *lin-52* mutants are resistant to a wide array of DNA-damage types, and the mutation alleviates the DNA-damage sensitivity of various mutants in single repair systems.

### DREAM inhibition boosts DNA-damage resistance in human cells

We next wondered whether inhibition of the highly conserved DREAM complex could provide a pharmacological approach to augment

---

**Fig. 4 | Mutations in the DREAM complex confer DNA-damage resistance against multiple damage types. a**, IR sensitivity dependent on HRR was tested in WT, *lin-52(n771)*, HRR-deficient *brc-1(tm1145); brd-1(dw1)*, NHEJ-deficient *cku-70(tm1524), lin-52(n771); brc-1(tm1145); brd-1(dw1), lin-52(n771); brc-1(tm1145); brd-1(dw1); cku-70(tm1524)* and *lin-52(n771); cku-70(tm1524)* worms. Data are shown as mean ± s.e.m., *n* = 4 independent experiments (*n* = 3 for *lin-52; cku-70*). Each independent experiment had three biological replicates. Two-tailed *t*-tests were used for statistical comparisons, and the most relevant comparisons are shown. **b**, IR sensitivity assay dependent on NHEJ repair in WT, *lin-52(n771)*, NHEJ-deficient *cku-70(tm1524)* and *lin-52(n771); cku-70(tm1524)* worms. The graph is representative of *n* = 3 biological replicates in 1 of 3 independent experiments, each of which had 3 biological replicates. Data are shown as mean ± s.d. of each larval stage. Two-tailed *t*-tests were used to compare the fraction of the larval stages of *lin-52* compared with WT, and *lin-52; cku-70* compared with *cku-70* (non-significant). **c**, Alkylation-damage assay of WT, *lin-52(n771)*,

alkylation-damage-sensitive *polh-1(lf31)* and double-mutant *lin-52(n771); polh-1(lf31)* worms. The graph is representative of *n* = 3 biological replicates in 1 of 3 independent experiments. Data are shown as mean ± s.d. of each larval stage. Two-tailed *t*-tests were used to compare the fraction of the larval stages of *lin-52* with WT, and *lin-52; polh-1* with *polh-1*. **d**, ICL assay of WT and *lin-52* mutants upon cisplatin treatment. Because cisplatin was diluted in DMF, worms were also given the maximum dose of DMF that was given for the cisplatin treatments as additional control. The graph is representative of *n* = 3 biological replicates in 1 of 3 independent experiments. Data are shown as mean ± s.d. of each larval stage. Two-tailed *t*-tests were used to compare the fraction of the larval stages of *lin-52* compared with WT. *P* > 0.05, not shown. **P* < 0.05, ***P* < 0.01, ****P* < 0.001, *****P* < 0.0001. Detailed *P* values and more comparisons can be found in Supplementary Table 13, including results from Fisher's exact test, which was used to analyze the distribution of the larval stages.

DNA-repair capacities in human cells. We analyzed ChIP–seq data from quiescent human cells[56] and searched for bound DNA-repair genes, following similar criteria as for the analyzed datasets from *C. elegans*[57]. We selected genes bound by at least LIN9, p130 and E2F4 simultaneously, where the binding occurred in 5′ between 0 and −1,000 bp of the TSS. Among the 328 gene promoters bound by DREAM (Supplementary Table 12), 67 genes are classified by GO as 'DNA repair' (Fig. 5a and Supplementary Table 12). Thus, the DREAM transcription repressor complex directly binds to DNA-repair gene promoters, indicating

that the DREAM-mediated DNA-repair gene regulation is conserved in *C. elegans* and humans.

In mammals, DREAM components not only form the DREAM repressor complex, but can also associate in other complexes that induce transcription[27–29]. We therefore used chemical inhibitors of the DYRK1A kinase, which phosphorylates LIN52, a modification required for the assembly of the DREAM complex, thus allowing its specific abrogation[30]. We employed two potent but distinct chemical inhibitors of the DYRK1A kinase: the beta-carboline alkaloid harmine, which has

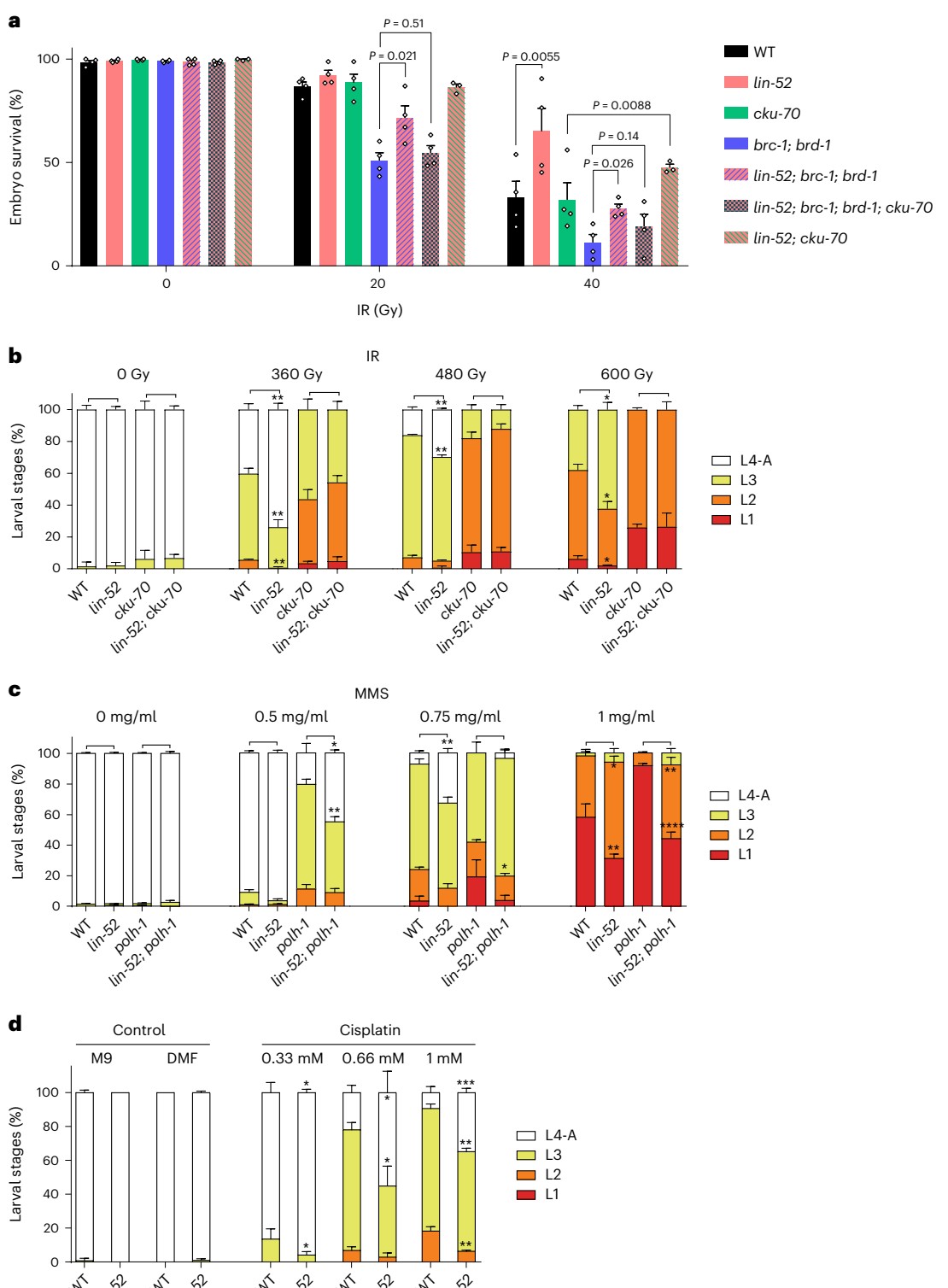

been widely used as specific DYRK1A inhibitor[31], and the benzothia-zole derivative INDY, which has been established as a highly selective DYRK1A inhibitor[32]. As the DREAM complex represses gene expression in G0 cells, we serum-starved confluent U2OS cells to obtain quiescent cell populations.

To confirm that the DYRK1A inhibitors abrogated DREAM-mediated gene repression, we performed RNA-seq analysis of quiescent cells treated with either harmine hydrochloride or INDY. Among the significantly upregulated genes (FDR-adjusted $P < 0.01$), all the known motifs bound by DREAM in human cells[56] were significantly overrepresented (Extended Data Fig. 8), thus substantiating that INDY and harmine treatment resulted in gene upregulation by inhibiting the DREAM complex. We plotted all the genes bound by the DREAM complex (Fig. 5a and Supplementary Table 12) that were significantly up- or downregulated upon harmine or INDY treatment (Fig. 5b, statistics in Supplementary Table 13), most of which were upregulated upon both treatments. Of the 67 DNA-repair genes bound by DREAM (Fig. 5a), 58 were upregulated upon harmine and 46 upon INDY treatment, and 45 were upregulated upon both treatments (Fig. 5b and Supplementary Table 12). These results indicate that the pharmacological inhibition of DYRK1A with harmine or INDY results in upregulation of DREAM target genes, including the majority of DREAM targets encoding DNA-repair genes.

To directly assess whether DYRK1A-inhibitor treatment could augment DNA-damage resistance, we exposed harmine- or INDY-treated quiescent cells to UV irradiation or the alkylating agent MMS. We measured the apoptotic response to the DNA damage using annexin V and 7-AAD analysis by flow cytometry (Fig. 5c–h). Both DYRK1A-inhibitor treatments resulted in a highly significant reduction in DNA-damage-induced apoptosis compared with mock-treated cells. In conclusion, pharmacological inhibition of the DREAM complex kinase DYRK1A increased the expression of DREAM-targeted DNA-repair genes and conferred resistance to distinct types of DNA damage, suggesting a highly conserved function of the DREAM complex in regulating DNA-repair capacities.

### Harmine treatment reduces retinal DNA damage

In humans, mutations in the NER pathway can lead to premature aging syndromes, such as Cockayne syndrome (CS), which is characterized by cachectic dwarfism, impaired development of the nervous system, pigmentary retinopathy (photoreceptor loss), cataracts, deafness and feeding difficulties[58,59]. In mice, mutations in *Ercc1*, which encodes an NER component, lead to a strong progeroid phenotype and premature death[60–62]. In order to test in vivo whether the inhibition of DYRK1A through harmine treatment could alleviate progeroid pathologies, we analyzed the loss of photoreceptor cells in the retina of *Ercc1*[−/−] mice, a hallmark of CS that can also be observed in NER-deficient mice, by performing TUNEL staining of the outer nuclear layer (ONL) of the retina[63].

*Ercc1*[−/−] and WT mice were harmine- or mock-treated intraperitoneally, and after 2 weeks of treatment, retinal degeneration was analyzed. *Ercc1*[−/−] mice showed increased TUNEL-positive apoptotic cells in the ONL compared with WT mice. The increased apoptosis levels in *Ercc1*[−/−] mice were significantly reduced upon harmine treatment (Fig. 5i,j).

Furthermore, we wondered whether the DNA damage present in the retinas of *Ercc1*[−/−] mice might be mitigated upon harmine treatment. Although the ONL shows very little γH2AX staining, even upon the induction of DNA damage, the cells in the inner nuclear layer (INL) typically show a stronger pan-nuclear γH2AX signal[64,65]. We quantified the γH2AX signal per nucleus in the INL in order to assess whether the treatment with harmine could reduce retinal DNA damage. The strong γH2AX signal in *Ercc1*[−/−] mice was significantly reduced upon harmine treatment (Fig. 5k,l).

In conclusion, treatment with harmine reduces the overall DNA-damage accumulation in *Ercc1*-deficient retinas and decreases the photoreceptor loss in this CS model.

## Discussion

Given the complexity of the repair mechanisms that respond to the distinct types of DNA lesions, it has remained elusive whether a mechanism exists that regulates the overall repair capacities of an organism. We uncovered that the *C. elegans* DREAM complex represses DNA-repair gene expression in somatic tissues, thus curbing their repair capacity and consequently limiting developmental growth, organismal health and lifespan upon DNA damage. Our data indicate that pharmacological targeting of the assembly of DREAM using DYRK1A inhibitors could be applied to augment DNA repair in human cells and in progeroid mice.

DNA repair in the germline is superior to that in the soma, with the germline preserving the genome, whereas somatic tissues accumulate mutations at increased rates with age[2–5]. Our data suggest that DREAM mutants confer enhanced germline-like DNA-repair capacities to somatic tissues. In *C. elegans*, it is not yet clear what mechanism keeps DREAM active in the soma and inactive in the germline. In mammals, the phosphorylation of LIN52 and the pocket proteins p130 and p107 is key to allowing the formation or disassembly of the complex, respectively. In *C. elegans*, the regulation might rely uniquely on the phosphorylation of the pocket protein LIN-35, which could disrupt the complex[66]. LIN-35 can be phosphorylated and inactivated by CDK-4 and the D-type cyclin CYD-1 (ref. [67]), which might inactivate DREAM in the germline. Understanding the mechanisms of DREAM inactivation might allow modulation of its activity to specifically enhance DNA repair.

The overall elevated DNA-repair gene expression also revealed a compensatory role between the distinct repair pathways. Abrogation of the DREAM complex enhances the removal of UV-induced DNA lesions, and even suppresses defects in GG-NER and TC-NER, but requires the presence of the NER machinery. These data suggest

---

**Fig. 5 | Inhibition of DREAM using DYRK1A inhibitors confers DNA-damage resistance in human cells and decreases DNA damage and apoptosis in the retinas of *Ercc1*-deficient mice. a**, Overlap between the DNA-repair genes in humans (GO database released 1 January 2021) and the genes bound by DREAM that were described in ref. [56]. A two-tailed Fisher's exact test was used for statistical analysis. **b**, FC comparison of DREAM target genes[56] whose expression levels are significantly changed upon harmine or INDY treatment (FDR < 0.01 in at least one of the datasets). DNA-repair genes are shown in orange. The upper-right quadrant of derepressed genes has a 2.39× over-enrichment. Detailed statistics are in Supplementary Table 13. **c,d,f,g**, Representative density plots of biological triplicates of U2OS cells, labeled with annexin V and 7-AAD, that were mock treated or that received harmine hydrochloride and/or UV (**c**), INDY and/or UV (**d**), harmine hydrochloride and/or MMS (**f**) or INDY and/or MMS (**g**). **e,h**, Percentage of apoptotic annexin-V-positive U2OS cells upon harmine (left) or INDY (right) treatment and UV (**e**) or MMS (**h**) treatment. Data are shown as mean ± s.d., *n* = 3 biological replicates. Two-tailed *t*-tests were used for statistical analysis between the populations under the same irradiation conditions.

**i**, Representative images of WT and *Ercc1*[−/−] retinas with TUNEL staining (green) and DAPI (blue). **j**, TUNEL-positive cells in the ONL from WT and *Ercc1*[−/−] mice upon treatment with harmine. *n* = 3 (2 male/1 female), 5 (3 male/2 female), 7 (4 male/3 female) and 7 (4 male/3 female) mice from left to right. Data are shown as mean ± s.e.m. For statistical analysis, two-tailed *t*-tests were used to compare groups that received or did not receive harmine treatment. **k**, Representative images of WT and *Ercc1*[−/−] retinas with γH2AX staining (red) and DAPI (blue). The intensity of the red channel has been equally increased in all images for visualization purposes. The INL of the retina is encircled by the dashed yellow line. **l**, γH2AX signal per nucleus in the INL of the retinas of WT and *Ercc1*[−/−] mice upon treatment with harmine. γH2AX signal per nucleus from an image stack per mouse is shown. *n* = 904, 889, 1,123 and 841 total nuclei, from left to right, from 5 (3 male/2 female) 5 (3 male/2 female), 6 (3 male/3 female) and 5 (3 male/2 female) imaged mice, respectively. Box midlines show the median, box limits show the top and bottom quartiles, and whiskers to 1.5 × interquartile range (IQR). For statistical analysis, two-tailed *t*-tests were used to compare groups that received or did not receive harmine treatment.

that, in DREAM mutants, activity of each NER sub-pathway is elevated, and their respective compensation is enhanced. DREAM mutants even suppress HRR deficiency through elevated NHEJ.

The suppression of *polh-1* mutants' sensitivity to ICLs and alkylating damage suggests that the consequence of DREAM abrogation might not necessarily be more error-prone repair. In the absence

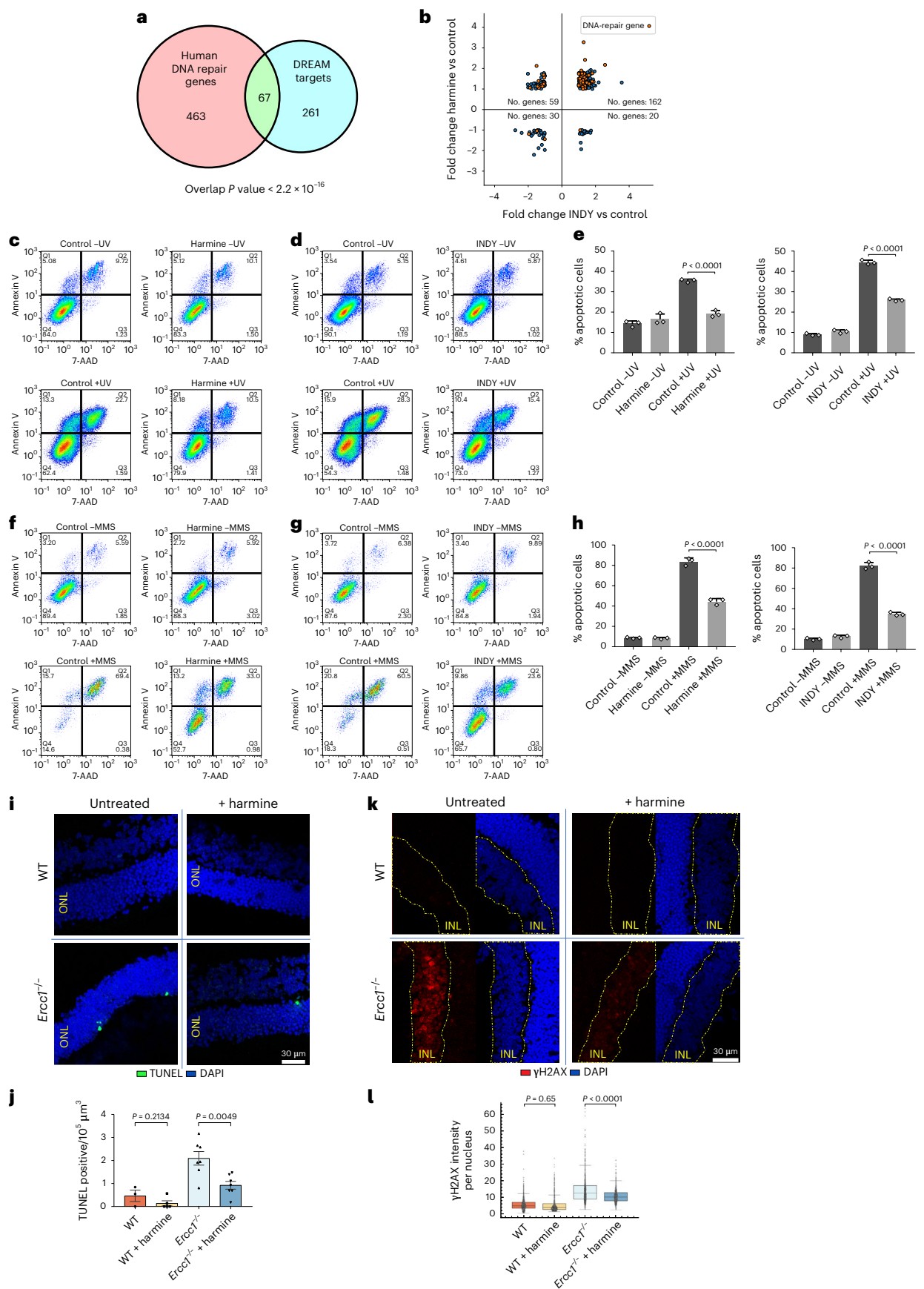

of DREAM, DNA-damage-driven developmental growth delay and aging are alleviated.

DREAM is a highly conserved transcription repressor complex that regulates the induction and maintenance of cellular quiescence by repressing cell cycle genes across multiple species[19,68]. Similar to the somatic cells in *C. elegans*, quiescent mammalian cells have limited DNA-repair capacities[18]. For example, quiescent hematopoietic stem cells (HSCs) and hair follicle stem cells use error-prone NHEJ[69], which has been shown to be accountable for the increased mutagenesis during HSC aging[70]. We established that pharmacological targeting of the DREAM complex could be applied to augment DNA-damage resistance in quiescent human cells. Considering that the conserved DREAM complex is highly active in post-mitotic cells and that neurons are particularly susceptible to DNA-repair defects, such as TC-NER defects in Cockayne syndrome, DSB-repair deficiencies in ataxia telangiectasia, or impaired SSB repair in cerebellar ataxia[71], targeting of the DREAM complex might provide new therapeutic avenues for a range of congenital DNA-repair deficiencies in humans. The alleviation of photoreceptor apoptosis upon harmine treatment in progeroid mice suggests that DREAM inhibition might indeed have therapeutic potential for DNA-damage-driven degenerative disorders. Intriguingly, the DYRK1A kinase is overexpressed in Down syndrome and involved in neurodegeneration in individuals with this condition, as well as in people with Alzheimer's, Parkinson's or Pick's disease[72-74]. It will be interesting to explore whether DREAM-complex inhibition could prevent DNA-damage driven conditions, such as stem-cell exhaustion, neurodegeneration and premature aging.

Our data establish the DREAM complex as a master regulator of DNA-repair-gene expression in somatic tissues and quiescent cells. Abrogation of the DREAM-mediated repression of DNA-repair genes elevates somatic repair capacities and enhances resistance to DNA damage. We propose that the DREAM complex restricts somatic DNA repair, and removal of this restriction confers germline-like DNA-repair capacities to the soma. Given the central role of nuclear genome stability in the aging process[75], the inhibition of the DREAM complex might provide a valuable intervention targeting DNA damage, which is a root cause of age-related diseases. Moreover, the suppression of various DNA-repair defects, such as GG- and TC-NER and HRR, suggests that the DREAM complex might also provide therapeutic opportunity in congenital DNA-repair deficiency syndromes that cause developmental growth failure and premature aging.

## Online content

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

## Methods

### *C. elegans* strains

All strains were cultured under standard conditions[78] and were always incubated at 20 °C during the experiments. The strains used were N2 (Bristol; WT):

DREAM: MT8839 *lin-52(n771) III*, MT10430 *lin-35(n745) I*, MT15107 *lin-53(n3368) I/hT2 [bli-4(e937) let-?(q782) qIs48] (I;III)*, MT8879 *dpl-1(n2994) II*, MT11147 *dpl-1(n3643) II*, JJ1549 *efl-1(se1) V*, BJS634 *dpl-1(n2994) II; lin-52(n771) III*.

SynMuv B and NuRD: RB951 *lin-13(ok838) III*, RB1789 *met-2(ok2307) III*, PFR40 *hpl-2(tm1489) III*, MT8189 *lin-15(n765) X*, MT14390 *let-418(n3536) V*.

NER: RB1801 *csb-1(ok2335) X*, FX03886 *xpc-1(tm3886) IV*, RB864 *xpa-1(ok698) I*, FX04539 *csa-1(tm4539) II*, BJS21 *xpc-1(tm3886) IV; csb-1(ok2335) X*, BJS631 *lin-52(n771) III; csb-1 (ok2335) X*, BJS629 *lin-52(n771) III; xpc-1 (tm3886) IV*, BJS630 *lin-52(n771) III; xpc-1 (tm3886) IV; csb-1 (ok2335) X*, BJS772 *xpa-1(ok698) I; lin-52(n771) III*, BJS825 *csa-1(tm4539) II; lin-52(n771) III*.

HRR: DW102 *brc-1 (tm1145); brd-1(dw1) III*, BJS890 *brc-1 (tm1145) III; brd-1(dw1); lin-52(n771) III*, BJS868 *lin-52(n771) III; brc-1(n771) III; brd-1(dw1) III; cku-70(tm1524) III*.

NHEJ: FX1524 *cku-70(tm1524) III*, BJS887 *lin-52(n771) III; cku-70(tm1524) III*.

MMS: XF132 *polh-1(lf31) III*, BJS722 *polh-1 (lf31) III; lin-52 (n771) III*.

### UV-irradiation assay during somatic development

The effects of UV-B in worm development were analyzed as previously described[79] after bleach synchronization[79]. UV-B irradiation was performed with a 310-nm PL-L 36W/UV-B UV6 bulb (Waldmann, 451436623-00005077). OP50 *Escherichia coli* was added to the plates, and worms were incubated for 48 h. Larval stages were determined using a dissecting microscope. For the strain MT15107 *lin-53(n3368) I/hT2 [bli-4(e937) let-?(q782) qIs48] (I;III)*, *lin-53(n3368)* homozygotes were distinguished using a fluorescence microscope (Leica M165 FC) and assessing the larval stage of worms that did not express green fluorescent protein.

### UV-irradiation assay for germline development

Synchronized late-L4 worms were irradiated with different doses of UV-B and allowed to recover for 24 h. Irradiated and mock-irradiated worms were transferred to a fresh seeded NGM plate to lay eggs for 4 h (5 worms per plate). Upon removal of the adults, the plates were incubated for 24 h, after which the number of eggs laid and percentage of eggs that survived and hatched were evaluated.

### NHEJ-dependent IR-sensitivity assay

As previously described[13,14], L1 worms repair DSBs mainly through NHEJ repair. To analyze worm sensitivity to IR in a NHEJ-dependent way, synchronized L1 worms were irradiated with different doses of IR using an IR-inducing cesium-137 source, and were left for 48 h to allow development. The different larval stages were determined using a dissecting microscope.

### HRR-dependent IR-sensitivity assay

Early embryos highly rely on HRR to repair DSBs[13,14]. To study the capacity of the different strains to tolerate IR-induced DSBs during embryogenesis, day-1 adults were left to lay eggs on seeded NGM plates for no longer than 1.5 h. Upon removal of the adult worms, early eggs were irradiated using an IR-inducing cesium-137 source. After 24 h, the percentage of surviving embryos that hatched was evaluated using a dissecting microscope.

### Alkylation-damage induction by using MMS

Synchronized L1 worms were incubated with different concentrations of MMS (Sigma, 129925) diluted in M9 buffer for 1 h. Worms were washed three times with M9 buffer, and plated in seeded NGM plates. After incubation for 48 h, worm development was evaluated with a dissecting microscope.

### ICL induction by using cisplatin

Synchronized L1 worms were exposed to different concentrations of cisplatin in dimethylformamide (DMF) diluted in M9 buffer or were mock-treated with DMF (Sigma, 227056)-diluted M9 buffer for 2 h. Worms were washed three times with M9 and incubated for 48 h in NGM plates, and the larval stages were quantified using a dissecting microscope.

### Lifespan assay

Synchronized day-1 adult worms were irradiated or mock-irradiated with UV-B light using a 310-nm PL-L 36W/UV-B UV6 bulb (Waldmann, 451436623-00005077) or with IR using an IR-inducing cesium-137 source, and then were placed on fresh OP50-seeded NGM plates. At the beginning of the experiment, the worms were transferred to new plates every other day to avoid progeny overgrowth. Worms presenting internal hatching or protruding or ruptured vulvas were censored and removed from the experiment, and worms were scored as dead when no movement or pumping was observed even upon physical stimulus. Lifespan curves were analyzed with Graphpad Prism 7.03 log-rank test.

### DNA-repair capacity assay in L1 worms

The quantification of DNA repair via immunostaining of CPDs and 6-4PPs of DNA samples in a slot blot was performed as has been described, with slight changes[79]. Bleach-synchronized L1 worms (at least 30,000 per plate) were irradiated with UV-B light and split in two groups, one to be immediately quick-frozen in liquid nitrogen, to serve as controls with unrepaired damage, and the other one was left in seeded plates for 24 h to allow for DNA repair to occur. After this, worms were washed 5 times, incubated for 2 h to permit the removal of intestinal bacteria, washed another 5 times and quick-frozen.

DNA extraction was performed using the Gentra Puregene Tissue Kit (Qiagen, 158667) and the protocol for DNA purification from tissue. The protocol was adapted to increase the volumes, but we still used a 1.5-ml Eppendorf tube to aid the supernatant extraction and pellet formation. That is, instead of the specified amounts, we used 500 μl cell lysis solution, 2.5 μl Puregene Proteinase K, 2.5 μl RNase A solution, 170 μl protein precipitation solution, 500 μl isopropanol and 500 μl 70% ethanol at the respective steps in the protocol. Cell lysis solution was directly added to the thawed sample, and an additional step with Proteinase K was performed. The DNA concentration was measured using the Qubit dsDNA HS Assay Kit (Invitrogen, Q32851). Serial dilutions of the DNA were denatured at 95 °C for 5 minutes (min) and transferred onto a Hybond nylon membrane (Amersham, RPN119B) using a Convertible Filtration Manifold System (Life Technologies, 11055). DNA crosslinking to the membrane was achieved by incubating the membrane at 80 °C for 2 h. The membrane was blocked for 30 min in 3% milk/phosphate-buffered saline (PBS)-T (0.1%) at room temperature. The membrane was incubated overnight at 4 °C with anti-CPD antibodies (Clone TDM-2, 1:10,000, Cosmo Bio, CAC-NM-DND-001) or anti-6-4 PP antibodies (Clone 64M-2, 1:3,000, Cosmo Bio, CAC-NM-DND-002), then washed three times with PBS-T (5 min at room temperature), and blocked for 30 min with 3% milk/PBS-T. The secondary antibody was a goat anti-mouse AffiniPure peroxidase-conjugated secondary antibody (1:10,000, Jackson Immuno Research, 115-035-174). Addition of the secondary antibody was followed by three washes in PBS-T and incubation with ECL Prime (Amersham, RPN2232). The DNA lesions were visualized by using a Hyperfilm ECL (Amersham, 28906836).

In order to quantify the total amount of DNA per sample, the membrane was incubated overnight at 4 °C in PBS with 1:10,000 SYBR Gold Nucleic Acid Stain (Invitrogen, S11494), then washed in PBS at room temperature and imaged using a BIO-RAD Gel Dox XR + Gel Documentation System (BIO-RAD, 1708195).

## Adult somatic and germline DNA-repair assay

Synchronized day-1 adult worms were irradiated or mock-irradiated with a 310-nm UV-B light Philips UV6 bulb in a Waldmann UV236B irradiation device. Half of the worms were left in seeded NGM plates for 60 h to allow DNA repair to occur, whereas the others were collected directly after the irradiation.

After irradiation or incubation, worms were picked and placed in a drop of M9 buffer on top of a HistoBond+ Adhesion Microscope Slide (Marienfeld, 0810461). Using a hypodermic needle, we cut the worms close to the head, which also releases one of the germline arms, and then placed a coverslip over the slide and kept it at −80 °C for at least 30 min. After this, the coverslip was removed quickly to perform freeze-cracking[80]. Worms were fixed in liquid methanol at −20 °C for 10 min, then washed for 5 min in PBS. Seventy microliters of 2 M HCl were added on top of the worms for 30 min at room temperature to denature the DNA. Slides were washed three times with PBS, and blocked with 70 μl of 20% fetal bovine serum (FBS) in PBS for 30 min at 37 °C. The slides were incubated with 70 μl of 1:10,000 anti-CPD antibodies (Clone TDM-2, 1:3,000, Cosmo Bio, CAC-NM-DND-001) in PBS containing 5% FBS, at 4 °C overnight in a humid chamber. After subjecting the slides to three PBS washes, 5 min each, 70 μl of secondary anti-mouse Alexa Fluor 488-conjugated antibody (1:300, Invitrogen, A21202) in 5% FBS PBS was added for 30 min at 37 °C. The slides were washed three times, 5 min each, and mounted using 5 μl of Fluoromount-G with DAPI (Invitrogen, 00495952). Images were obtained using a SP8 Confocal Microscope by Leica using LAS X 3.5.7 software.

## Image quantification of CPDs

Image stacks of the heads and germlines of adult worms were analyzed using the analysis software Imaris 9.9 (Oxford Instruments). Nuclei in the area anterior to the pharyngeal-intestinal valve and germlines were determined by using DAPI staining and setting a threshold of size and intensity. False-positive nuclei (due to bacteria in the pharynx) were manually discarded. The CPD signal was quantified using the maximum spherical volume fitting inside each of the nuclei. Owing to variable background signal in the germlines, background intensity was subtracted in these samples.

## Cell culture and treatments

U2OS (ATCC, HTB-96) were cultured in DMEM, high-glucose GlutaMAX supplement, pyruvate (Thermo Fisher Scientific, 31966047) with 10% fetal bovine serum (FBS; Biochrom, S0615) and 1% penicillin–streptomycin (Thermo Fisher Scientific, 15140112). Cells were kept at 37 °C in a 5% $CO_2$ incubator (Binder). Cell dissociation was performed with Accutase (Sigma, A6964). To promote quiescence, cells were cultivated in FBS-free medium for 48 h before genotoxic treatment. After 24 h of culture in FBS-free medium, cells were mock treated or received harmine hydrochloride (diluted in water) or INDY (diluted in DMSO) (Sigma, SMB00461 and SML1011) at 10 or 25 μM, respectively. Before the genotoxic treatment, cells were washed with FBS-free medium. For the UV treatment, medium was removed from the plates and cells were irradiated using 254-nm UV-C light Philips UV6 bulbs with 2 mJ/cm². The MMS treatment was performed by adding MMS at 2 mM for 2 h, followed by 3 washes with FBS-free medium. Then, FBS-free medium was added. Quantification via flow cytometry of cell death and apoptosis was performed 24 h after genotoxic treatment.

## Flow cytometry analysis

Collected cells were incubated in annexin V binding buffer (BioLegend, 422201) with Pacific Blue annexin V (BioLegend, 640917) and 7-AAD (Thermo Fisher Scientific, 00699350) at 4 °C for 15 min. Cells were measured using a MACSQuant VYB (Miltenyi Biotec) using MACSQuantify software 2.13.0 and analyzed using FlowJo v10.7.1 (BD). The gating strategy can be found in Supplementary Figure 1.

## Animal handling

All animals were maintained in their breeding cages on a 12-h light/dark cycle. Mice were kept on a regular diet and had access to water ad libitum. Body weight was measured weekly. Animals were housed in a temperature- (18–23 °C) and humidity-controlled (40–60%), pathogen-free animal facility at the Institute of Molecular Biology and Biotechnology (IMBB), which operates in compliance with the 'Animal Welfare Act' of the Greek government, using the 'Guide for the Care and Use of Laboratory Animals' as its standard. All experiments were performed under the Animal license 6ΛΤΑ7ΛΚ-ΚΚΘ, issued by the Veterinary Medicine Directorate of Greek Republic.

## Mice experiments

Male and female FVB/nj:C57BL/6j *Ercc1*⁻/⁻ and their respective control WT mice[81], on the third day after birth (postnatal day P3), were injected intraperitoneally 3 times per week with 10 mg/kg body weight of harmine hydrochloride (SMB00461, Sigma) diluted in 0.9% sodium chloride. Mice were euthanized at postnatal day P15 for retina tissue isolation. Tissues were embedded in optimal cutting temperature (OCT) compound, cryosectioned and stained using the *in situ* cell death detection kit (TUNEL staining) (11684817910, Roche), according to the manufacturer's instructions.

For the immunostaining experiments against γH2AX (Millipore, 05–636), retina slices were fixed in 4% formaldehyde in 1× PBS for 10 min at room temperature, permeabilized with 0.5% Triton X-100 in 1× PBS for 10 min, on ice, and blocked with 1% BSA in 1× PBS for 1.5 h at room temperature. After overnight incubation with the primary antibody (1:12,000, in 1% BSA/1× PBS, 4 °C), a secondary fluorescent antibody was added (goat anti-mouse IgG-Alexa Fluor 555, 1:2,000, Invitrogen, A-21422) and DAPI (1:20,000, Thermo Fisher Scientific, 62247) was used for nuclear counterstaining.

Samples were visualized with an SP8 TCS laser scanning confocal microscope (Leica). The detection of nuclei and signal intensity from retinas was performed utilizing Imaris 9.9 (Oxford Instruments).

## RNA extraction for RNA-seq and qPCR experiments

For the qPCR and RNA-seq of L1 worms, around 10,000 (qPCR) or 40,000 (RNA-seq) bleach-synchronized L1 worms in triplicates (qPCR) or quadruplicates (RNA-seq) per strain and condition were placed in seeded NGM plates for 3 h. They were mock-treated or UV-B irradiated, and left for 6 h to allow the DNA-damage-related transcriptional changes to take place. Worms were collected and washed three times with M9 buffer, and the pellet was placed in a tube containing 1 ml TRIzol (Invitrogen, 15596018) and 1 mm zirconia/silica beads (Biospec Products, 11079110z).

To extract the RNA, worms were disrupted with a Precellys24 (Bertin Instruments, P000669-PR240-A), and the RNA isolation was performed by using the RNeasy Mini Kit (QIAGEN, 74106) following the manufacturer's specifications, except we used 1-bromo-3-chloropropane (Sigma, B9673) instead of chloroform. The RNA was quantified using NanoDrop 8000 (Thermo Fisher Scientific, ND-8000-GL).

RNA extraction from U2OS cells was performed after 24 h of harmine or INDY treatment of cells that had been starved for a total of 48 h by using the RNeasy Mini Kit (QIAGEN, 74106), following the manufacturer's specifications. Cells were disrupted with RLT buffer and homogenized with QIAshredder spin columns (QIAGEN, 79656).

## qPCR

Reverse transcription to form complementary DNA (cDNA) was performed using Superscript III (Invitrogen, 18080044). The obtained cDNA was used to perform qPCR by using SYBR Green I (Sigma, S9460) and Platinum Taq polymerase (Invitrogen, 10966034) in a BIO-RAD CFX96 real-time PCR machine (BIO-RAD, 1855196). The analysis of the results was performed by using the comparative $C_T$ method[82].

All *C. elegans* qPCR experiments were done in biological triplicates, and the data were normalized to three housekeeping genes. qPCR $C_T$ values were obtained using Bio-Rad CFX Manager 3.0.

**C. elegans qPCR primers.** Primers used for PCR were as follows:

| Housekeeping genes: | Forward primer | Reverse primer |
|---|---|---|
| Y45F10D.4 | 5′-CGAGAACCCGC GAAATGTCGGA-3′ | 5′-CGGTTGCCAGGGAAG ATGAGGC-3′ |
| eif-3.C | 5′-ACACTTGACGAG CCCACCGAC-3′ | 5′-TGCCGCTCGTTCCTTC CTGG-3′ |
| vha-6 | 5′-CTGCTATGTCAA TCTCGG-3′ | 5′-CGGTTACAAATTTCAA CTCC-3′ |
| **Genes of interest:** | | |
| parp-1 | 5′-AGCGAATGAAGA AACAATCCGA-3′ | 5′-ACTAGGCGTTCGATT ACTTGTG-3′ |
| polh-1 | 5′-AGAAATATCGCGA CGCTAGC-3′ | 5′-GTAGGTAATAGCAGC CTGCA-3′ |
| polk-1 | 5′-GAGATACTGATGG AGAATCTTGAG-3′ | 5′-AGTAGTTGGATGTG CTCAGC-3′ |
| mus-101 | 5′-TCGAAAGCCAT ATACGATGAACC-3′ | 5′-ACAAGAACGGGAG TACTAGAGAC-3′ |
| exo-3 | 5′-GGAGGAGACGT TTAAGAACTACAC-3′ | 5′-TAGATCACTGGCTT CTTCTCGT-3′ |
| lig-1 | 5′-TGATCAAGGCTG TTGCTAAAGC-3′ | 5′-AGCCTCAATTCCTT GACATGC-3′ |
| atm-1 | 5′-GCGAAGTTCTTA CACCTCGAC-3′ | 5′-AGTTCGACACATTCT TCAGCA-3′ |

## RNA-seq

A triplicate of RNA samples from *lin-52* mutant and WT L1 worms were rRNA-depleted using Ribo-Zero Plus rRNA Depletion Kit (Illumina, 20037135) and sequenced using a Hiseq4000 (Illumina) with PE75 read length. For RNA quality control, the RNA integrity number was ≥9.4 for all samples. RNA-seq data were processed through the QuickNGS pipeline[83], Ensembl version 85. Reads were mapped to the *C. elegans* genome using Tophat[84] (version 2.0.10) and abundance estimation was done using with Cufflinks[85] (Version 2.1.1). DESeq2 (ref. [76]) was used for differential gene expression analysis.

The human RNA-seq data were processed with Salmon-1.1 (ref. [86]) against a decoy-aware transcriptome (gencode.v37 transcripts and the GRCh38.primary_assembly genome) with the following parameters: −validateMappings −gcBias −seqBias. The output was imported and summarized to the gene-level with tximport (1.14.2)[87], and differential gene analysis was done with edgeR (3.28.1)[88].

## Proteomics

WT and *lin-52(n771)* L1 worms were plated in OP50-containing NGM plates and left to feed for 9 h. Worms were collected and washed 5 times with M9 buffer to remove the OP50, and 8 M urea buffer mixed in 50 mM TEAB with 1× Protease Inhibitor cocktail (Roche) was added to the sample before quick freezing.

Chromatin was degraded using a Bioruptor (Diagenode) for 10 min with cycles of 30/30 seconds. Upon centrifugation, the concentration of protein in the supernatant was calculated using Qubit Protein Assay Kit (Thermo Fisher Scientific). Twenty-five micrograms of protein per sample were transferred to a new tube, dithiothreitol was added to a concentration of 5 mM followed by vortexing and incubation at 25 °C for 1 h. Chloroacetamide was added to a final concentration of 40 mM, and the samples were incubated at room temperature for 30 min. Protein digestion with lysyl endopeptidase was done at an enzyme:substrate ratio of 1:75 and incubated at 25 °C for 4 h. Samples were diluted with 50 mM TEAB to reach a urea concentration of 2 M, and trypsin protein digestion was performed by

adding trypsin at an enzyme:substrate ratio of 1:75; the samples were kept at 25 °C overnight.

After protein digestion, SDB RP StageTip purification was performed[89]. The protein samples were then analyzed utilizing liquid chromatography–mass spectrometry by the CECAD Proteomics Facility on a Q Exactive mass spectrometer that was coupled to an EASY nLC 1000 (Thermo Fisher Scientific). The differential protein levels were obtained by CECAD's Proteomics Facility. Briefly, a predicted spectrum library was generated using the Prosit webserver[90], and data were processed using DIA-NN 1.7.16 (ref. [91]) and imported into Perseus 1.5.5.0 (ref. [91]) for analysis.

Four replicates per strain and condition were used, each containing around 20,000 worms.

## Datasets

The list of 211 genes belonging to the GO term 'Cellular response to DNA damage stimulus' was obtained by using data from the GO Consortium[92,93] (database released on 8 October 2019) (Supplementary Table 1). GO analysis was performed using the PANTHER 15.0 over-representation test and Venn Diagrams were created using Venn Diagram Plotter 1.5 and GIMP 2.10.12.

Gene IDs from previously published datasets were updated to current databases. Duplicated or dead IDs were eliminated accordingly. Overlap analysis were done by using Fisher's exact test in R v3.6.3 (ref. [94]). Gene set enrichment analysis (GSEA) was done in R v3.6.3 (ref. [94]) with the GSEA function of clusterProfiler v3.14.3 (ref. [95]) and the parameter settings minGSSize = 3, maxGSSize = 5000, and nPerm = 20000. To calculate the adjusted *P* values for the GSEA results, statsmodels[96] v0.11.1 multipletests methods with the parameter method = 'fdr_bh' or method = 'bonferroni' in Python 3.6 (ref. [97]) was used.

## Promoter analysis: *C. elegans*

The set of 211 DDR genes was used as input for the findMotifs function of HOMER-4.11-2 (ref. [98]) with the parameters -len 8,10 -start −1000 -end 0. Wormbase IDs were converted to the sequence name with WormBase's SimpleMine[99]. These identifiers were searched in the 'worm.description' file of HOMER to gain the corresponding RefSeq IDs. The *P* values were calculated with the hypergeometric tests function in scipy(1.5.1). HOMER's seq2profile function[98] was used to convert the previously reported CDE + CHR DREAM complex motif[35] with one mismatch and three random base pairs in between to a motif file usable by HOMER with the following parameter: seq2profile.pl BSSSSSNNNTTYRAA 1 (ref. [35]). The constructed motif was searched with the findMotifs.pl -find function for the 211 DDR genes with the parameters -start −1000 -end 0. The background enrichment of the motif was calculated for all 20,174 protein-coding genes with a RefSeq ID included in the worm.description file of HOMER. The *P* values were calculated with the hypergeometric tests function in scipy (1.5.1)[100].

## Promoter analysis:human

Homer's seq2profile function was used to convert previously reported DREAM complex motifs[56], with no allowed mismatch, with the following parameters:

- seq2profile.pl TTTSSCGS 0
- seq2profile.pl VVCGGAAGNB 0
- seq2profile.pl BNBVNTGACGY 0
- seq2profile.pl CWCGYG 0

The motifs were searched with the findMotifs.pl -find function with the parameters -start −1000 -end 0 for the up and downregulated genes, after harmine and INDY treatments, respectively, with an FDR cutoff of 0.01.

The background enrichment of the motif was calculated for all protein-coding genes included in the homer.description file of HOMER.

The *P* values were calculated with the hypergeometric test function in scipy(1.5.1)[100] and Python's statsmodels (0.11.1)[96] was used to calculate the Benjamini–Hochberg FDR.

## Statistics and reproducibility

In *C. elegans*, development growth assays and egg-laying assays were performed a minimum of 3 times, each of which included 3 biological replicates per condition with an average of around 40–50 individuals per sample. Slot blots were performed at least three times. Flow cytometry assays were done at least three times, each having three biological replicates. For experiments from which data were obtained from single individuals, such as mice experiments, worm imaging studies, and lifespan assays of worms, the sample sizes are indicated. All attempts at replication were successful.

The sample sizes have been well established in similar experiments in other scientific publications (refs. [9,13,39,79], among others). The statistical analysis performed in each experiment can be found in the figure legend. Two-tailed *t*-tests were done in Microsoft Excel 2019 and GraphPad Prism 7.03. log-rank, Mann–Whitney and two-way ANOVA tests for the germline image quantification tests and unpaired *t*-tests with Welch's correction were done in GraphPad Prism 7.03. Two-way ANOVA for the quantification of worm heads was done with Python's pingouin v0.3.6. Two-tailed Fisher's exact tests were done in RStudio 1.2.5019. *C. elegans* experiments were not randomized was not applied because the group allocation was guided on the basis of the genotype of the respective mutant worms. Worms of a given genotype were nevertheless randomly selected from large strain populations for each experiment without any preconditioning. In mice experiments, allocation was random.

Blinding was generally not applied, as the experiments were carried out under highly standardized and predefined conditions to avoid investigator-induced bias. Developmental assays upon DNA damage with small observed effects were performed blinded to exclude any bias. This affects the developmental growth upon IR, MMS and cisplatin treatment.

## Starvation assay

A pool of synchronized starving *lin-52* mutant and WT L1 worms was maintained in M9 buffer rolling at 20 °C. From this pool, around 30 worms were transferred to seeded NGM plates over consecutive days, and the number of L1 worms per plate was counted. After 48–72 h in the seeded plates, the number of worms that recovered and survived the ongoing starvation was evaluated. A biological triplicate for each strain was used for each timepoint. The experiment was discontinued after 14 days because all worms from days 13 and 14 had died. Each plate per condition, replicate and day had an average of 30 worms.

**Total-egg-hatching assay.** Single synchronized day 1 adult worms were transferred to seeded NGM plates and left to lay eggs for 24 h. Each day, the worms were transferred to a new seeded NGM plate, until egg laying stopped for all individuals. The total number of eggs laid and hatching/surviving eggs were evaluated 24 h after the removal of the adult from the plate. Twelve adult WT and *lin-52* mutant worms were used. Internal hatching or exploding worms were excluded from the day the event occurred onwards.

**Motility assay.** Synchronized day 1 adult *lin-52* mutant or WT worms were UV-irradiated or mock-treated and incubated at 20 °C for 72 h. Next, 30 worms were transferred to unseeded small NGM plates and left for 30 min to avoid worms accumulating in areas with food, promote movement and facilitate image analysis. To obtain video footage, the plates containing the worms were left under the microscope light without the lid for 30 seconds to allow the worms to get used to the conditions. Thirty-second videos were taken by using a Zeiss Axio Zoom V.16 and Zeiss ZEN 2.3 pro software. Worm footage was analyzed using the plugin wrMTrck in ImageJ 1.53q.

**EdU-incorporation assay in L1 and adult worms.** Thymidine-deficient *E. coli* (strain MG1693) were grown in M9 containing 1% glucose, 1 mM $MgSO_4$, 1.25 µg/ml vitamin B1, 0.5 µM thymidine and 20 µM 5-ethynyl-2′-deoxyuridine (EdU) at 37 °C overnight in darkness. These bacteria were used to seed M9-agar plates (M9 with 1.2% agar and 0.6% agarose) and were left to incubate overnight at room temperature.

Synchronized L1 or adult *lin-52* mutant and WT worms were UV-irradiated or mock-treated and transferred to the plates containing the EdU-labeled MG1693 *E. coli*. Worms were collected after 6 h, 12 h and 24 h for L1 worms, and after 24 h for adults, washed three times in M9 buffer and transferred to fixing buffer (1× egg buffer with 0.1% Tween and 3% PFA). Fixed worms were placed on top of a Histo-Bond+ Adhesion Microscope Slides (Marienfeld, 0810461), adult worms were cut open, and a coverslip was placed above the worms and slight pressure was applied. Next, the slides were placed on dry ice to allow freeze-cracking[80]. We used the Click-iT EdU imaging kit (Invitrogen, C10337), following the manufacturer's instructions, for the preparation of the Click-iT reaction cocktail. Upon washing the worms 3 times for 5 min in PBS, 50 µl of reaction cocktail was added to the slides, followed by incubation for 30 min at room temperature in darkness. Slides were washed once in 3% BSA in PBS and mounted using 5 µl of Fluoromount-G with DAPI (Invitrogen, 00495952). Images were obtained using a SP8 Confocal Microscope by Leica using LAS X 3.5.7 software. Positive EdU nuclei were counted manually from the obtained image stacks.

## Reporting summary

Further information on research design is available in the Nature Portfolio Reporting Summary linked to this article.

## Data availability

The *C. elegans* proteomics data used in this study have been deposited to the ProteomeXchange Consortium via the PRIDE partner repository with identifier PXD033836.

The *C. elegans* RNA-seq data used in this study are available from Gene Expression Omnibus (GEO; http://www.ncbi.nlm.nih.gov/geo) under the accession number GSE152235.

The human RNA-seq data are available under the accession number GSE168401.

gencode-v37 transcripts and GRCh38.primary_assembly genome can be accessed at:

https://ftp.ebi.ac.uk/pub/databases/gencode/Gencode_human/release_37/

Ensembl version 85 data can be accessed at:

https://ftp.ensembl.org/pub/release-85/gtf/caenorhabditis_elegans/

Data from refs. [35,44,46,47,56] were re-analyzed. Source data are provided with this paper.

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

## Acknowledgements

We thank the Cologne Center for Genomics (CCG), the CECAD bioinformatics and proteomics facility, and the Regional Computing Center of the University of Cologne (RRZK) for providing computing time on the DFG-funded High Performance Computing (HPC) system CHEOPS, as well as support. Worm strains were provided by the National Bioresource Project (supported by The Ministry of Education, Culture, Sports, Science and Technology, Japan), the Caenorhabditis Genetics Center (funded by the NIH National Center for Research Resources, US), and the *C. elegans* Gene Knockout Project at the Oklahoma Medical Research Foundation (part of the International *C. elegans* Gene Knockout Consortium). A. B. received a scholarship for doctoral candidates from the Fundación 'la Caixa' and the German Academic Exchange Service (DAAD). D. H. M. is a member of the Cologne Graduate School of Ageing Research. B. S. acknowledges funding from the Deutsche Forschungsgemeinschaft (FOR 5504 project 496650118, SCHU 2494/3-1, SCHU 2494/7-1, SCHU 2494/10-1, SCHU 2494/11-1, SCHU 2494/15-1, CECAD EXC 2030 – 390661388, SFB 829, KFO 329, and GRK2407), the Deutsche Krebshilfe (70114555) and the John Templeton Foundation Grant (61734).

## Author contributions

A. B. conceived and designed the study, performed the experiments and analyzed the data, G. S. performed experiments and analyzed data, A. B. and D. H. M. performed the bioinformatic analyses. G. C. and K. S. performed the mouse experiments that they designed with G. A. G. and analyzed together with A. B. B. S. coordinated the project and together with A. B. designed the study. A. B. and B. S. wrote the paper.

## Funding

## Competing interests

B. S. and G. A. G. are co-founders of Agevio Therapeutics, Inc. All other authors declare no competing interests.

## Additional information

**Extended data** is available for this paper at https://doi.org/10.1038/s41594-023-00942-8.

**Correspondence and requests for materials** should be addressed to Björn Schumacher.

**Peer review information** Dimitris Typas was the primary editor on this article and managed its editorial process and peer review in collaboration with the rest of the editorial team. Peer reviewer reports are available.

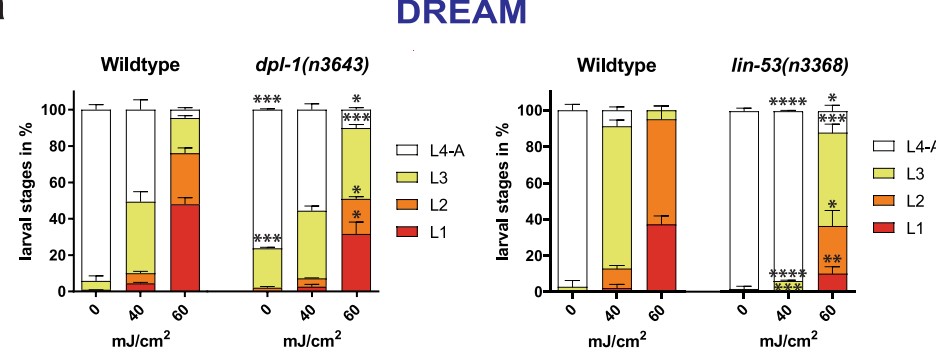

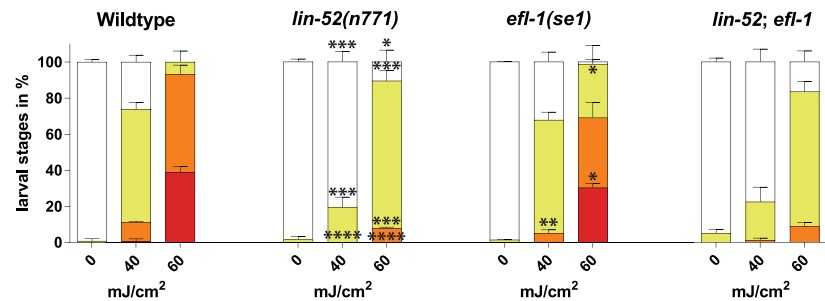

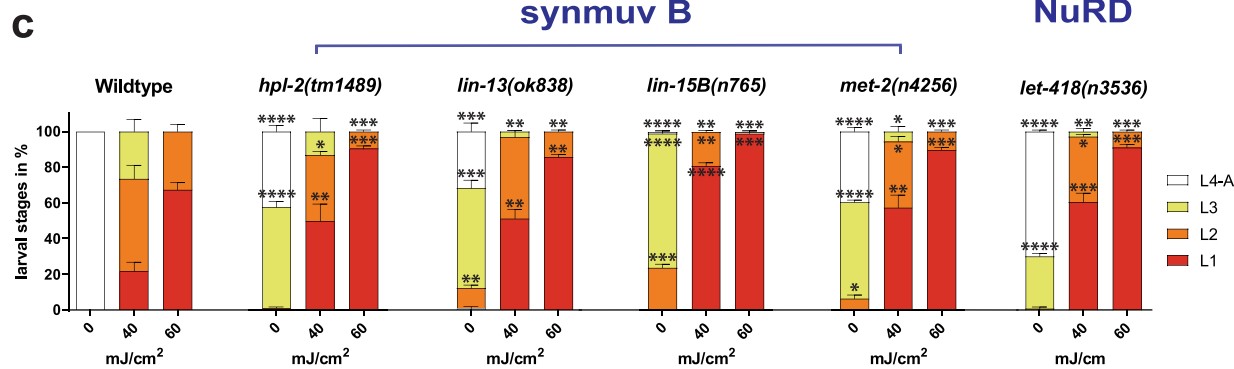

**Extended Data Fig. 1 | Mutations of components of the DREAM complex but not of other synMuv B or NuRD components confer resistance to UV-induced DNA damage during development. a**, UV-irradiation assay during somatic development of WT, *dpl-1(n3643)* and *lin-53(n3368) I/hT2 [bli-4(e937) let-?(q782) qIs48]*. Larval stages were determined 48 h post-irradiation. Homozygous *lin-53(n3368)* were identified by inspection with a fluorescence microscope (non-GFP population). Representative graph showing *n* = three biological replicates from one out of at least three independent experiments. Mean +/− SD of each larval stage. Two-tailed *t*-test between the fraction of each larval stage of a mutant compared to WT in the same treatment condition. **b**, UV-irradiation assay during somatic development of WT, *lin-52(n771)*, *efl-1 (se1)* and *lin-52(n771); efl-1 (se1)* mutants. Representative graph showing *n* = three biological replicates from one

out of three independent experiments. Mean +/− SD of each larval stage. Two-tailed *t*-test between the fraction of each larval stage of a mutant compared to WT in the same treatment condition for the single mutants, and the comparison of the double mutant and *lin-52(n771)* are shown. **c**, UV-irradiation assay during somatic development of WT, *hpl-2(tm1489)*, *lin-13(ok838)*, *lin-15(n765)*, *met-2(n4256)* and *let-418(n3536)* mutants. Representative graph showing *n* = three biological replicates from one out of at least two independent experiments. Mean +/− SD of each larval stage. Two-tailed *t*-test between the fraction of each larval stage of a mutant compared to WT in the same treatment condition. If *P* > 0.05, statistics not shown. **P* < 0.05, ***P* < 0.01, ****P* < 0.001, *****P* < 0.0001. Detailed *P* values and Fisher's exact tests to compare the population distribution can be found in the Supplementary Table 13.

**a**

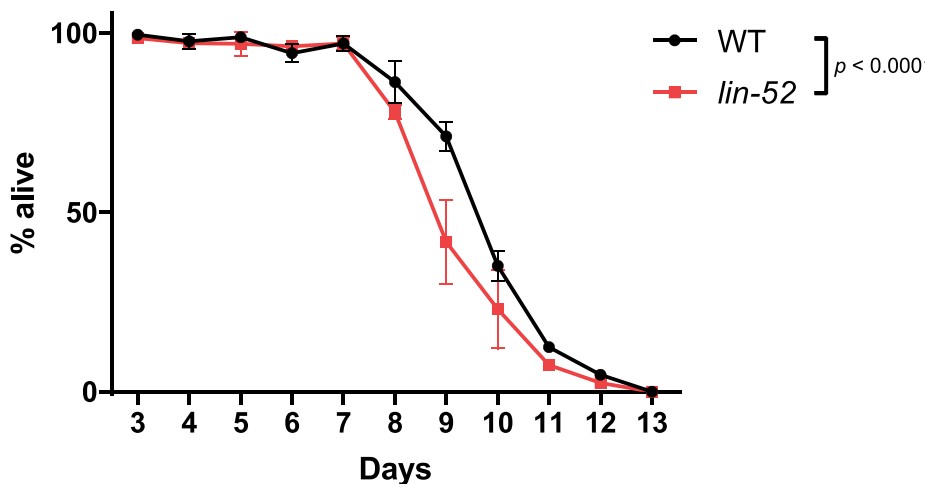

**b**

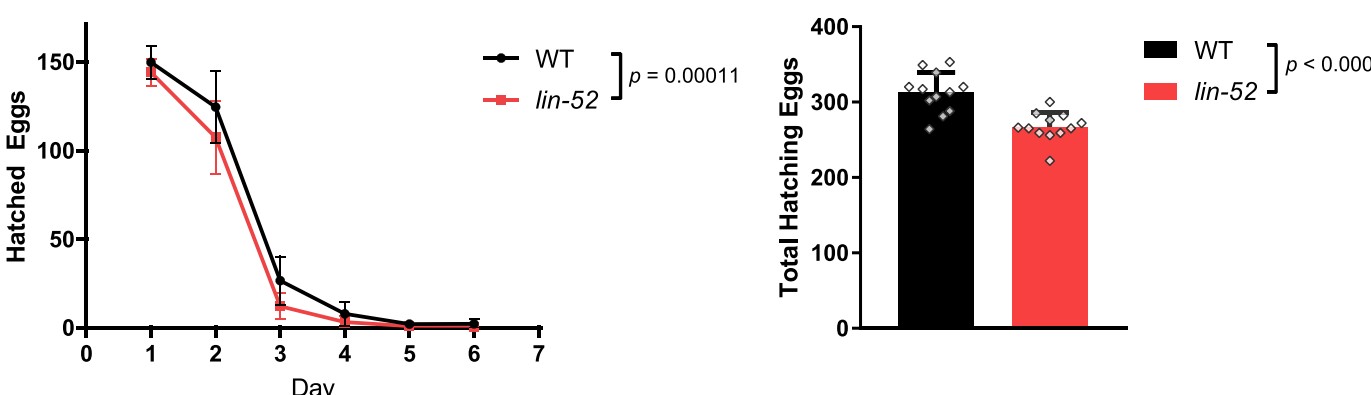

**Extended Data Fig. 2 | lin-52 mutant worms show a slight starvation sensitivity and reduced fecundity. a,** Starved L1 larvae of WT and *lin-52(n771)* were plated in seeded plates each day and allowed to recover from starvation. The mean +/− SD percentage of worms surviving the starvation period compared to the total of worms per plate is presented, *n* = three biological replicates per day for each strain. The *P* value of the strain variable from two-way ANOVA is shown. **b,** Average of hatching eggs laid by adult *lin-52(n771)* and WT worms per day (left) or during their reproductive lifespan (right). *n* = 12 individual worms per strain were used, mean +/− SD is shown. The *P* value of the strain variable from two-way ANOVA is shown (left) or unpaired two-tailed *t*-test (right) between the strain.

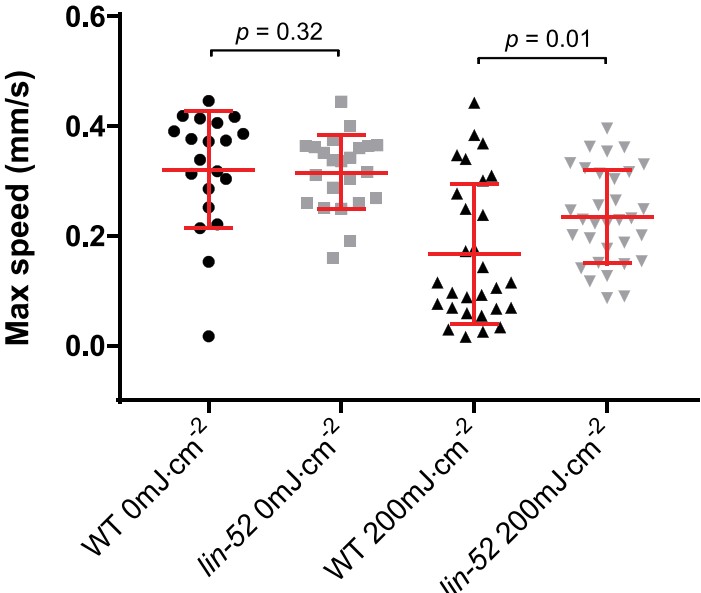

**Extended Data Fig. 3 | lin-52 mutant worms retained motility upon DNA damage induction.** *lin-52(n771)* and WT day 1 adult worms were irradiated or mock-treated with UV-B and their motility was measured after 72 h. Maximum speed per individual is presented (mean +/− SD in red). Two-tailed Mann-Whitney test was used. From left to right, *n* = 20, 23, 29, 32.

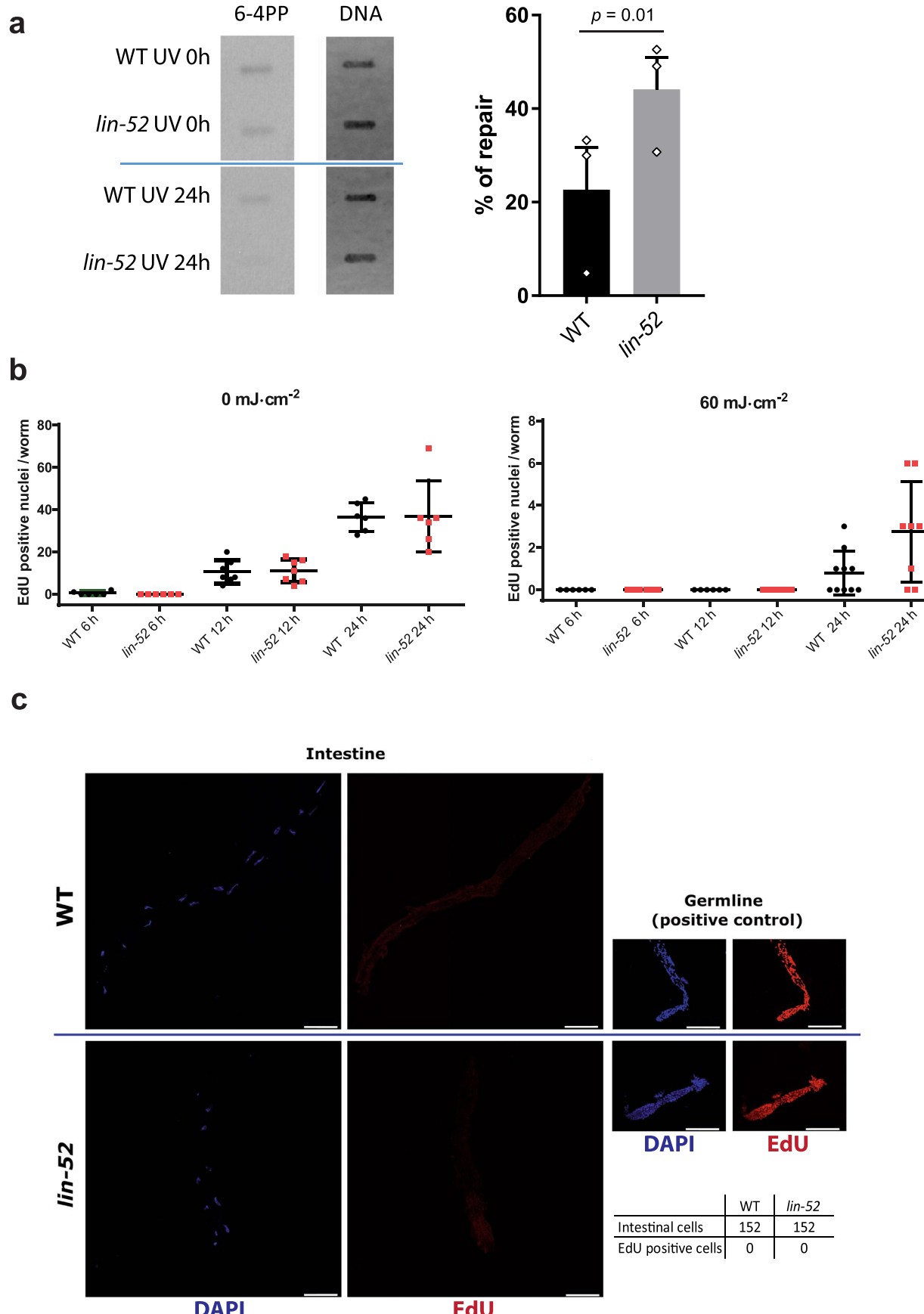

**Extended Data Fig. 4 | See next page for caption.**

**Extended Data Fig. 4 | lin-52 mutant worms show improved UV damage repair capacity without having additional replication events compared to WT. a**, Representative slot blot out of three independent experiments labelled with antibodies against 6-4PPs and SYBR™ Gold for DNA staining. DNA samples were collected from irradiated worms (WT and *lin-52(n771)*) right after or 24 h after 60 mJ·cm² UV-B irradiation. Graph show the percentage of damage repair of WT and *lin-52* mutant worms after 24 h compared to 0 h (*n* = 3, mean +/− SEM). Two-tailed *t*-test was used to compare the repair with WT worms. *$P$ < 0.05. Uncropped blot in Source Data ED Fig. 4. **b**, Mock-treated (left) or UV-B irradiated (right) WT and *lin-52(n771)* L1-stage worms were EdU labelled and imaged at different timepoints. Representative graphs of two independent experiments. The number of replication events per individual worm are presented (average with SD). Unpaired *t*-test with Welch's correction was applied. No significant differences were found. From left to right, *n* = 6, 6, 7, 7, 6, 6 // 6, 6, 6, 7, 10, 8 worms. **c**, Day 1 adult *lin-52* mutant and WT worms were fed with EdU containing bacteria for three days. Worms were cut open and intestines and germlines were analyzed using confocal microscopy to find EdU positive nuclei. *n* = 152 intestinal cells of WT and *lin-52* mutants were counted. All observed germlines were EdU positive. Scale bar, 75 μm.

**a**

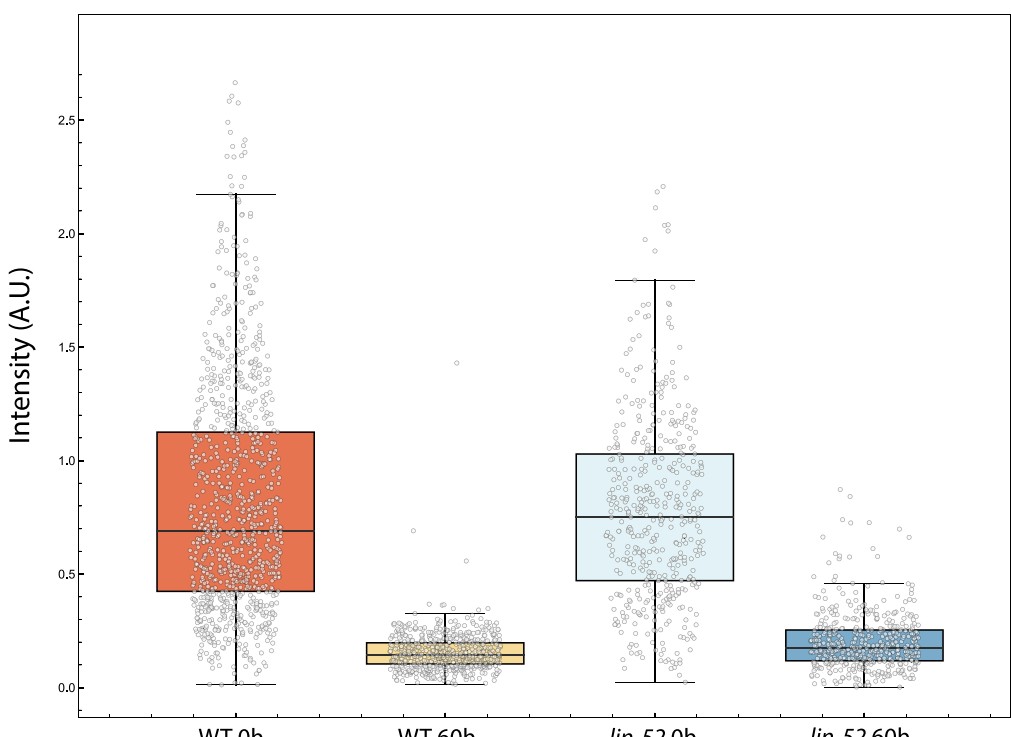

| Source of Variation | % of total variation | p-value |
|---|---|---|
| Interaction | 0.2254 | 0.0019 |
| TIME | 38.99 | <0.0001 |
| GENOTYPE | 0.0005303 | 0.8804 |

**b**

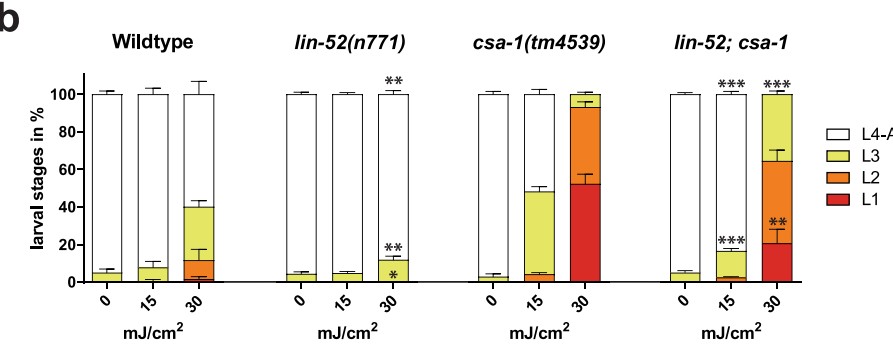

**c**

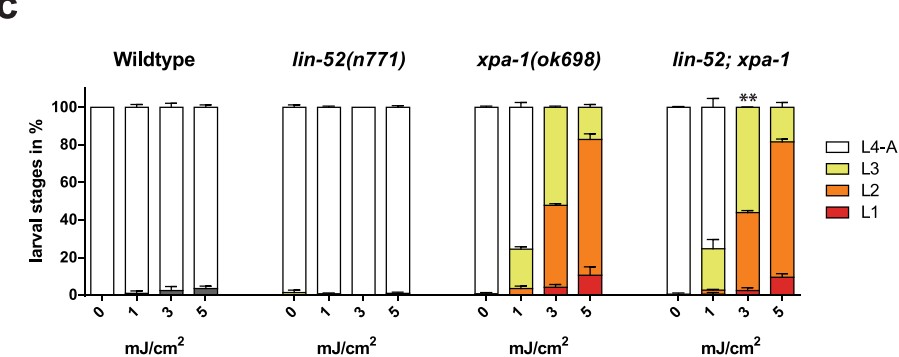

**Extended Data Fig. 5 | See next page for caption.**

**Extended Data Fig. 5 | Germline DNA repair is similar between WT and lin-52 mutant worms and mutant lin-52 alleviates the UV sensitivity of csa-1 but not of xpa-1 mutants. a**, Quantification of CPD nuclei signal intensity in the germline of adult worms irradiated and collected immediately or 60 h after irradiation. The number of nuclei quantified was $n$ = 962, 462, 479, 494 from left to right from 3–5 germlines per strain and condition. The $y$ axis shows the CPD intensity normalized to nuclear DAPI with subtracted background. Box depicts the median with top and bottom quartiles, whiskers to 1.5 IQR. 2-way ANOVA analysis shows a negligible effect of the genotype. **b**, UV-irradiation assay during somatic development of WT, *lin-52(n771)*, *csa-1(tm4539)* and *lin-52(n771); csa-1(tm4539)* mutants. Representative graph showing $n$ = three biological replicates from one out of three independent experiments. Mean +/− SD of each larval stage. Two-tailed *t*-tests between the fraction of each larval stage of a *lin-52; csa-1* compared to *csa-1* and *lin-52* compared to WT in the same treatment condition are presented. **c**, UV-irradiation assay during somatic development of WT, *lin-52(n771)*, *xpa-1(ok698)* and the *lin-52(n771); xpa-1(ok698)* mutants. Representative graph showing $n$ = three biological replicates from one out of two independent experiments. Mean +/− SD of each larval stage. Two-tailed *t*-tests between the fraction of each larval stage of a *lin-52; xpa-1* compared to *xpa-1* and *lin-52* compared to WT in the same treatment condition are presented. If $P > 0.05$, statistics not shown. *$P < 0.05$, **$P < 0.01$, ***$P < 0.001$. Detailed $P$ values and all comparisons against WT and Fisher's exact tests to compare the population distribution can be found in the Supplementary Table 13.

**a**

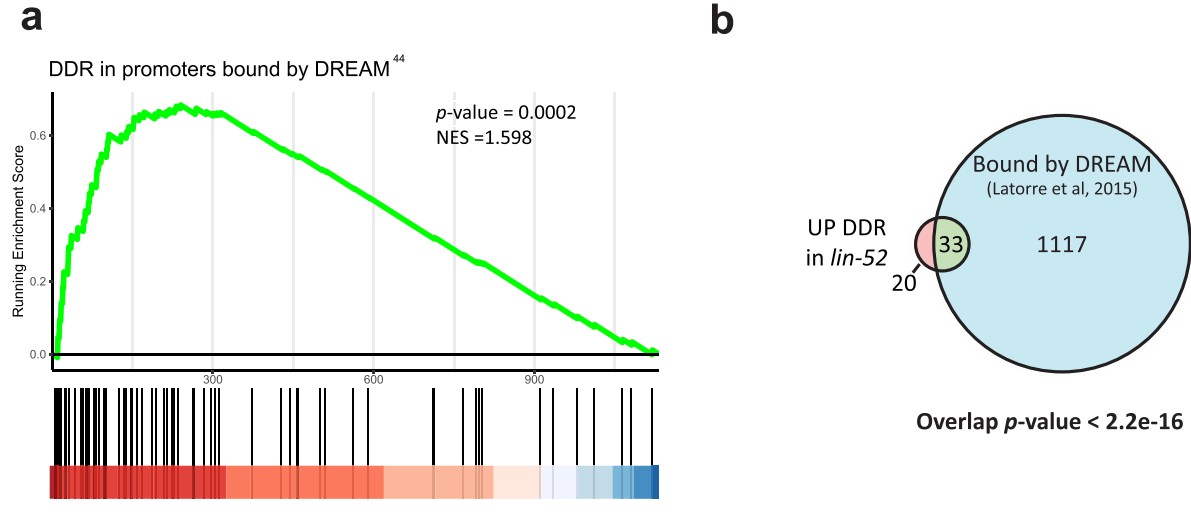

**b**

**c**

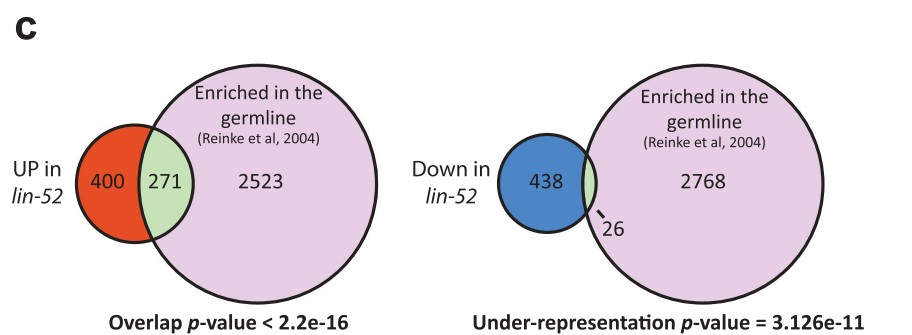

**d**

**e**

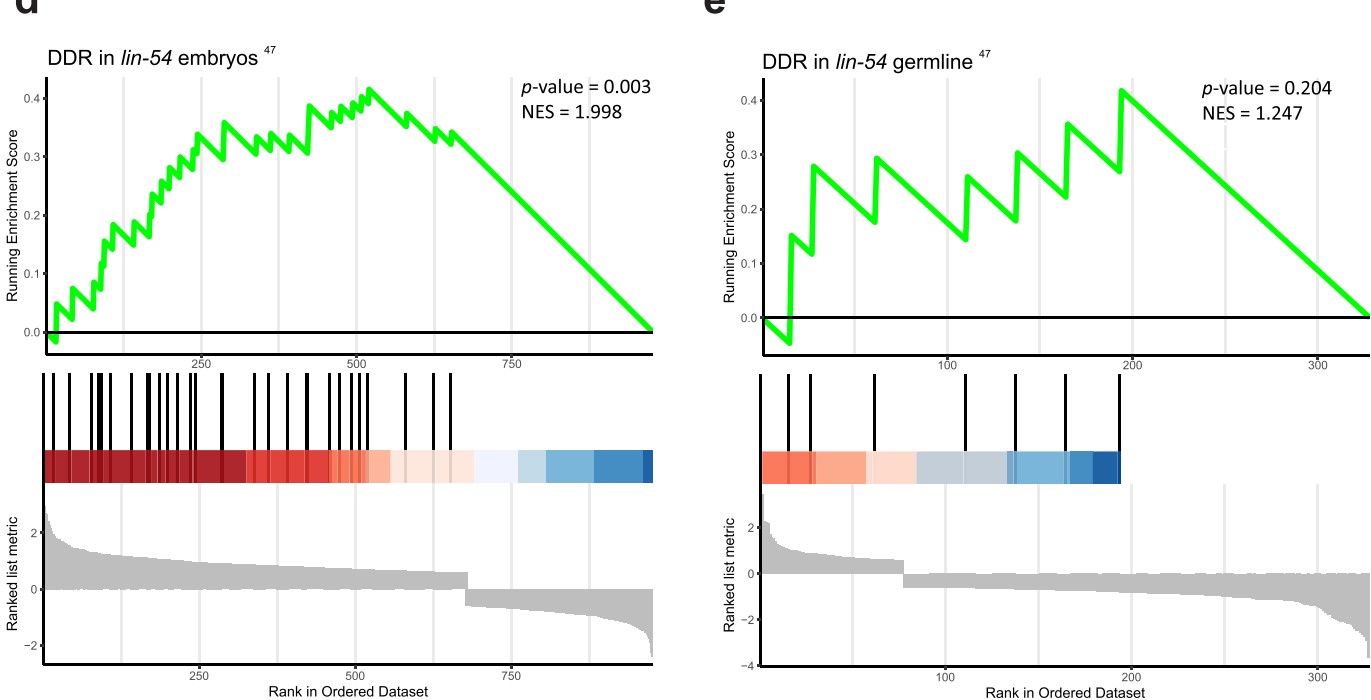

**Extended Data Fig. 6 | See next page for caption.**

**Article**                                                    https://doi.org/10.1038/s41594-023-00942-8

**Extended Data Fig. 6 | DREAM binds DDR genes and DREAM´s mediated DDR repression is not occurring in the germline. a**, GSEA of all the DDR genes in the RNA-seq of *lin-52(n771)* with the genes which promoters are bound by DREAM upon analysis of reference[44]. 63 out of the 211 DDR genes are found in both and used for the analysis. In red, DDR genes bound by DREAM and upregulated, in blue, downregulated. **b**, Overlap between the genes bound by DREAM and all the DDR genes (Supplementary Table 1) (11 not found). Re-analysis of reference [44]. Overlap *P* value calculated by performing Fisher's exact test. **c**, Overlap between a dataset of germline-enriched genes[46] and all the genes significantly up-regulated in *lin-52* mutants compared to WT worms (*p*-adj < 0.05) (left) and the genes found down-regulated in *lin-52* mutants compared to WT worms (*p*-adj < 0.05) (right).

Overlap *P* value was calculated by Fisher's exact test. **d**, GSEA of the DDR genes within a microarray of *lin-54* mutant embryos from reference[47]. 30 out of the 211 DDR genes are found in the *lin-54* embryo dataset consisting of 977 genes and used for the analysis. In red, DDR genes upregulated in *lin-54* mutant embryos, in blue, downregulated. **e**, GSEA of the DDR genes within a microarray of *lin-54* mutant germlines from reference[47]. 7 out of the 211 DDR genes are found in the *lin-54* germline dataset consisting of 328 genes and used for the analysis. In red, DDR genes upregulated in *lin-54* mutant germlines, in blue, downregulated. Two-sided Fisher's exact test is shown for overlap analysis. GSEA statistics done as described in[77].

**a**

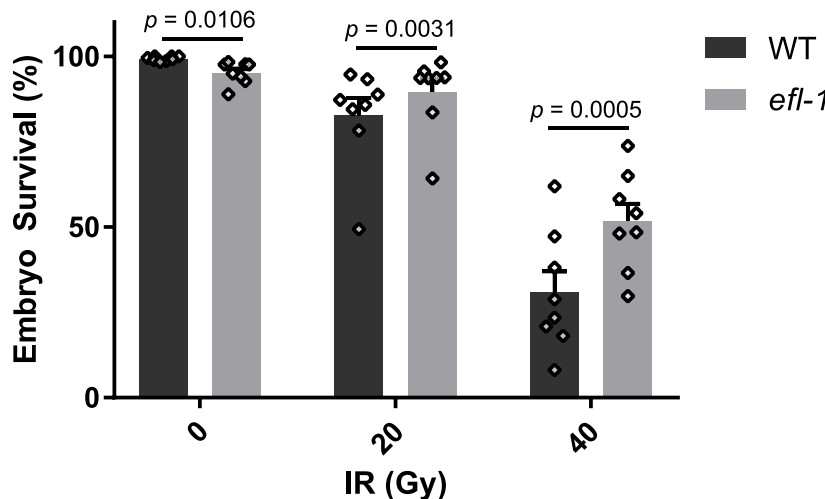

**b**

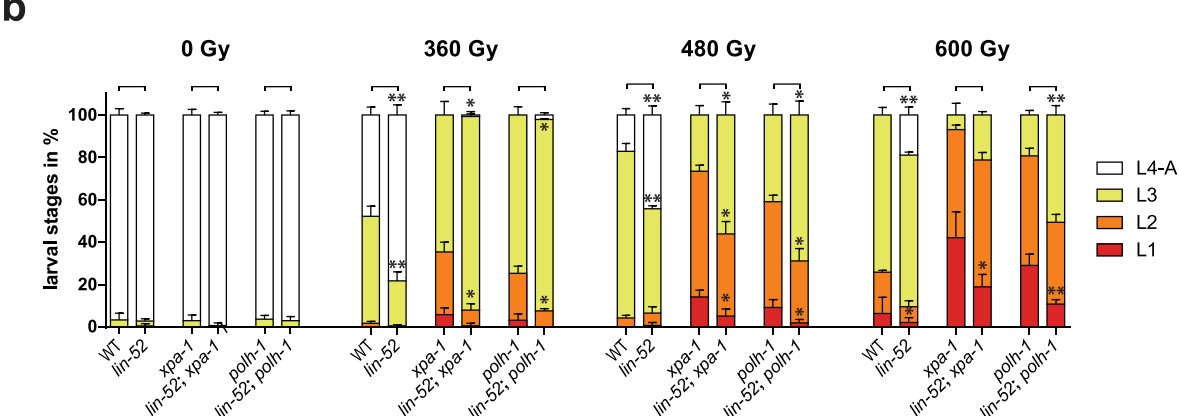

**c**

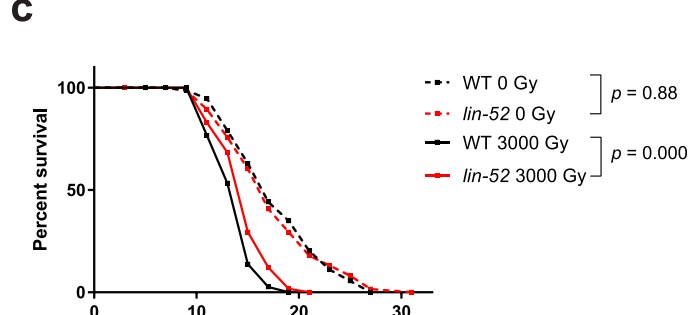 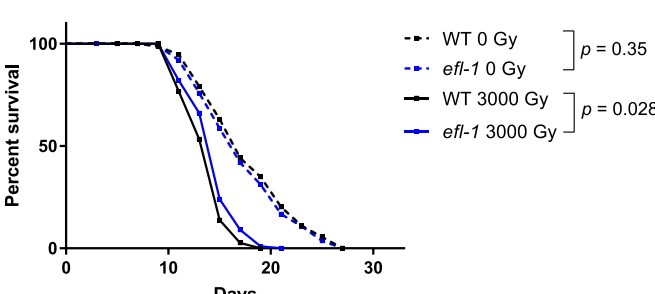

**Extended Data Fig. 7 | DREAM complex mutants confer DSB damage resistance and can partially alleviate the sensitivity of DNA repair deficient strains. a**, HRR-dependent IR sensitivity assay in WT and *efl-1(se1)* mutant. Graph shows the mean of 8 independent experiments, with SEM, each of them with three biological replicates. Two-tailed *t*-tests between the fraction of surviving embryos within the treatment are presented. **b**, NHEJ-dependent IR sensitivity assay in WT, *lin-52(n771)*, *xpa-1(ok698)*, *polh-1(lf31)* and the double mutants *lin-52(n771);xpa-1(ok698)* and *lin-52(n771);polh-1(lf31)*. Representative graph of one out of three independent experiments, each with *n* = 3 biological replicates. Mean +/− SD of each larval stage. Two-tailed *t*-tests between the fraction of

the larval stages of *lin-52* compared to WT, *lin-52; xpa-1* compared to *xpa-1* and *lin-52; polh-1* compared to *polh-1* are presented (showed statistics indicated with upper bracket). **c**, Lifespan assay upon IR (control and 1000 Gy treatment) of WT, *lin-52(n771)* and *efl-1(se1)* mutant worms. log-rang (Mantel-Cox) test was performed to compare the lifespan of the DREAM mutants and WT in the same conditions. Without IR and with IR respectively, *n* = 62 and 118 for WT, 63 and 116 for *lin-52(n771)* and 85 and 100 for *efl-1(se1)*. For **b**, *P* > 0.05, statistics not shown. *\*P* < 0.05, \*\**P* < 0.01. The precise *P* values and all comparisons to WT worms and Fisher's exact tests to compare the population distribution can be found in the Supplementary Table 13.

| Motif | Treatment | % of UP Genes | % of BG | FC | *p*-value | *q*-value |
|---|---|---|---|---|---|---|
| TTTGGCCGG (E2F) | Harmine | 16.83 | 12.85 | 1.31 | 1.1e-11 | 1.83e-11 |
| TTTGGCCGG (E2F) | INDY | 15.86 | 12.85 | 1.24 | 5.29e-11 | 7.55e-11 |
| AACGGAAGAC (NRF2) | Harmine | 25.03 | 18.41 | 1.36 | <1e-12 | <1e-12 |
| AACGGAAGAC (NRF2) | INDY | 25.39 | 18.41 | 1.38 | <1e-12 | <1e-12 |
| CACAATGACGT (CREB) | Harmine | 17.99 | 14.92 | 1.21 | 3.52e-7 | 4.4e-7 |
| CACAATGACGT (CREB) | INDY | 18.48 | 14.92 | 1.24 | 4.16e-12 | 1.04e-11 |
| CACGTG (n-MYC) | Harmine | 60.02 | 49.04 | 1.22 | 1.09e-11 | 1.83e-11 |
| CACGTG (n-MYC) | INDY | 56.66 | 49.04 | 1.16 | <1e-12 | <1e-12 |

**Extended Data Fig. 8 | Analysis of DREAM recognition sites in promoters of genes significantly upregulated upon harmine hydrochloride or INDY treatment in U2OS cells.** The 4 previously reported promoter motifs bound by DREAM[56] corresponding to E2F, NRF2, CREB and n-MYC consensus motifs from top to bottom, were searched in the gene-sets of significantly (FDR < 0.01) upregulated genes by either harmine or INDY. All four motifs in both treatments show a highly significant overrepresentation. The *P* value was calculated with two-sided hypergeometric tests and the *q* value shows the Benjamini–Hochberg adjusted *P* values. BG, background; FC, fold change.

| | |
|---|---|

# Reporting Summary

## Statistics

For all statistical analyses, confirm that the following items are present in the figure legend, table legend, main text, or Methods section.

| n/a | Confirmed | |
|---|---|---|
| ☐ | ☒ | The exact sample size (*n*) for each experimental group/condition, given as a discrete number and unit of measurement |
| ☐ | ☒ | A statement on whether measurements were taken from distinct samples or whether the same sample was measured repeatedly |
| ☐ | ☒ | The statistical test(s) used AND whether they are one- or two-sided<br>*Only common tests should be described solely by name; describe more complex techniques in the Methods section.* |
| ☐ | ☒ | A description of all covariates tested |
| ☐ | ☒ | A description of any assumptions or corrections, such as tests of normality and adjustment for multiple comparisons |
| ☐ | ☒ | A full description of the statistical parameters including central tendency (e.g. means) or other basic estimates (e.g. regression coefficient) AND variation (e.g. standard deviation) or associated estimates of uncertainty (e.g. confidence intervals) |
| ☐ | ☒ | For null hypothesis testing, the test statistic (e.g. *F*, *t*, *r*) with confidence intervals, effect sizes, degrees of freedom and *P* value noted<br>*Give P values as exact values whenever suitable.* |
| ☒ | ☐ | For Bayesian analysis, information on the choice of priors and Markov chain Monte Carlo settings |
| ☒ | ☐ | For hierarchical and complex designs, identification of the appropriate level for tests and full reporting of outcomes |
| ☒ | ☐ | Estimates of effect sizes (e.g. Cohen's *d*, Pearson's *r*), indicating how they were calculated |

*Our web collection on statistics for biologists contains articles on many of the points above.*

## Software and code

Policy information about availability of computer code

| Data collection | qPCR CT values were obtained using Bio-Rad CFX Manager 3.0.<br>Videos for worm movement assessment were captured using ZEN 2.3 Pro software (Zeiss).<br>Other microscopy images were captured using LAS X 3.5.7 Life Science microscope software (Leica).<br>Flow Cytometry data was obtained using MACSQuantify software 2.13.0. |
|---|---|
| Data analysis | C. elegans RNA-seq data were processed through the QuickNGS pipeline, Ensembl version 85. The reads were mapped using Tophat (2.0.10) and abundance estimation was done with Cufflinks (version 2.1.1). DESeq2 was used for differential expression analysis.<br>Human RNA-seq data was processed with Salmon-1.1 and the output was summarized to the gene-level with tximport (1.14.2) and the differential gene analysis was done with edgeR (3.28.1).<br>C. elegans promoter and human promoter analysis was done using Homer's seq2profile function. p-values were calculated with the hypergeometric test function in scipy (1.5.1) and Python's statsmodels (0.11.1) was used to calculate the Benjamini-Hochberg FDR.<br>C. elegans proteomic data was analyzed by generating a prediced spectrum library using the Prosit webserver, and data was processed using DIA-NN 1.7.16 and imported to Perseus 1.5.5.0 for analysis.<br>Gene ontology analysis was performed using PANTHER 15.0 overrepresentation test.<br>Overlap analysis were done by using Fisher's exact test was done in R and gene set enrichment analysis (GSEA) was done in R v3.6.3 with the GSEA function of clusterProfiler v3.14.3. To calculate the adjusted p-values for the GSEA results, statsmodels v0.11.1 multipletests methods with the parameter method='fdr_bh' or method='bonferroni' in Python 3.6 was used.<br>2-tailed t-tests and the comparative Ct method for qPCRs were done using Microsoft Excel 2019.<br>Flow cytometry analysis was performed in FlowJo v10.7.1.<br>ANOVA of CPD signal data was performed with Python's pingouin v0.3.6.<br>Survival curves in C. elegans were analysed with Graphpad prism 7.03. |

Microscopy image analysis was done using Imaris 9.9 (Oxford Instruments).
Venn Diagrams were done using Venn Diagram Plotter 1.5 and GIMP 2.10.12.
Motility of worms was analyzed using the plugin wrMTrck in ImageJ 1.53q.

For manuscripts utilizing custom algorithms or software that are central to the research but not yet described in published literature, software must be made available to editors and reviewers. We strongly encourage code deposition in a community repository (e.g. GitHub). See the Nature Portfolio guidelines for submitting code & software for further information.

## Data

Policy information about availability of data

All manuscripts must include a data availability statement. This statement should provide the following information, where applicable:
- Accession codes, unique identifiers, or web links for publicly available datasets
- A description of any restrictions on data availability
- For clinical datasets or third party data, please ensure that the statement adheres to our policy

The C. elegans RNA-seq data used in this study are available from Gene Expression Omnibus (GEO; http://www.ncbi.nlm.nih.goz/geo) with the accession number GSE152235 and secure token yvqvewuetzkjrsl.
The C. elegans proteomics data used in this study is available via ProteomeXchange with identifier PXD033836.
The human RNA-seq data is available with the accession number GSE168401 and secure token wjehcacaflkxrip.
gencode-v37 transcripts and GRCh38.primary_assembly genome can be accessed from:
https://ftp.ebi.ac.uk/pub/databases/gencode/Gencode_human/release_37/
Ensembl version 85 data can be accessed from:
https://ftp.ensembl.org/pub/release-85/gtf/caenorhabditis_elegans/

Data from the following articles was re-analised, of which the availability is dependent on the journal and the institution from which it is accessed:
- Goetsch PD, Garrigues JM, Strome S. Loss of the Caenorhabditis elegans pocket protein LIN-35 reveals MuvB's innate function as the repressor of DREAM target genes. PLoS Genet. 2017 Nov 1;13(11):e1007088. doi: 10.1371/journal.pgen.1007088. PMID: 29091720; PMCID: PMC5683655.
- Reinke V, Gil IS, Ward S, Kazmer K. Genome-wide germline-enriched and sex-biased expression profiles in Caenorhabditis elegans. Development. 2004 Jan;131(2):311-23. doi: 10.1242/dev.00914. Epub 2003 Dec 10. PMID: 14668411.
- Tabuchi TM, Deplancke B, Osato N, Zhu LJ, Barrasa MI, Harrison MM, Horvitz HR, Walhout AJ, Hagstrom KA. Chromosome-biased binding and gene regulation by the Caenorhabditis elegans DRM complex. PLoS Genet. 2011 May;7(5):e1002074. doi: 10.1371/journal.pgen.1002074. Epub 2011 May 12. PMID: 21589891; PMCID: PMC3093354.
- Latorre I, Chesney MA, Garrigues JM, Stempor P, Appert A, Francesconi M, Strome S, Ahringer J. The DREAM complex promotes gene body H2A.Z for target repression. Genes Dev. 2015 Mar 1;29(5):495-500. doi: 10.1101/gad.255810.114. PMID: 25737279; PMCID: PMC4358402.
- Litovchick L, Sadasivam S, Florens L, Zhu X, Swanson SK, Velmurugan S, Chen R, Washburn MP, Liu XS, DeCaprio JA. Evolutionarily conserved multisubunit RBL2/p130 and E2F4 protein complex represses human cell cycle-dependent genes in quiescence. Mol Cell. 2007 May 25;26(4):539-51. doi: 10.1016/j.molcel.2007.04.015. PMID: 17531812.

## Human research participants

Policy information about studies involving human research participants and Sex and Gender in Research.

| Reporting on sex and gender | N/A |
| --- | --- |
| Population characteristics | N/A |
| Recruitment | N/A |
| Ethics oversight | N/A |

Note that full information on the approval of the study protocol must also be provided in the manuscript.

# Field-specific reporting

Please select the one below that is the best fit for your research. If you are not sure, read the appropriate sections before making your selection.

☒ Life sciences        ☐ Behavioural & social sciences        ☐ Ecological, evolutionary & environmental sciences

For a reference copy of the document with all sections, see nature.com/documents/nr-reporting-summary-flat.pdf

# Life sciences study design

All studies must disclose on these points even when the disclosure is negative.

| Sample size | Sample size are well stablished and commonly used on similar experiments in other scientific publications (Among others, Bianco et al, 2018, Rieckher et al 2017, Mueller et al 2014, Johnson NM et al 2013, Umansky et al, 2022) |
| --- | --- |

| Data exclusions | No data were excluded |
|---|---|
| Replication | In C. elegans, development growth assays and egg laying assays were performed a minimum of three independent times, each included 3 biological replicates per genotype and condition with an average around 40-50 individuals per sample and replicate across experiments. Slot blots were performed at least three independent times. Flow cytometry assays were done at least three independent times, each having 3 biological replicates within the experiment. Experiments from which single individual data was obtained, such as mice experiments, imaged worms, worms in lifespan assays, have the sample size indicated for each experiment. All attempts at replication were successful.<br>All performed experiments in mice are reported in the summary and all results were consistent. |
| Randomization | Randomization was not applied because the group allocation was guided based on the genotype of the respective mutant worms. Worms of a given genotype were nevertheless randomly selected from large strain populations for each experiment without any preconditioning. In mice experiments, allocation was random. |
| Blinding | Blinding was generally not applied as the experiments were carried out under highly standardized and predefined conditions such that an investigator-induced bias can be excluded. Developmental assays upon DNA damage with small observed effects were performed blinded to avoid any bias from the strain. This affects all replicates and repetitions of the developmental growth upon IR, MMS and Cisplatin. |

# Reporting for specific materials, systems and methods

We require information from authors about some types of materials, experimental systems and methods used in many studies. Here, indicate whether each material, system or method listed is relevant to your study. If you are not sure if a list item applies to your research, read the appropriate section before selecting a response.

## Materials & experimental systems

| n/a | Involved in the study |
|---|---|
| ☐ | ☒ Antibodies |
| ☐ | ☒ Eukaryotic cell lines |
| ☒ | ☐ Palaeontology and archaeology |
| ☐ | ☒ Animals and other organisms |
| ☒ | ☐ Clinical data |
| ☒ | ☐ Dual use research of concern |

## Methods

| n/a | Involved in the study |
|---|---|
| ☒ | ☐ ChIP-seq |
| ☐ | ☒ Flow cytometry |
| ☒ | ☐ MRI-based neuroimaging |

## Antibodies

| Antibodies used | + Primary antibodies used for immunofluorescence staining:<br>- Anti-CPD (Clone TDM-2, Supplier Cosmo Bio, CAC-NM-DND-001)<br>- Anti-6,4-PP (Clone 64M-2, Supplier Cosmo Bio, CAC-NM-DND-002)<br>- Anti phospho Histone H2A.X (Ser139) (Clone JBW301, Supplier Millipore, 05-636)<br>+ Secondary antibodies:<br>- AffiniPure Peroxidase-conjugated secondary antibody (Polyclonal, Jackson Immuno Research, 115-035-174)<br>- Anti-mouse IgG Alexa Fluor 555 (Polyclonal, Invitrogen, A-21422)<br>- Anti-mouse Alexa Fluor 488 (Polyclonal, Invitrogen, A-21202) |
|---|---|
| Validation | - Anti-CPD (Clone TDM-2, Supplier Cosmo Bio, CAC-NM-DND-001): https://www.cosmobiousa.com/products/anti-cpds-mab-clone-tdm-2<br>- Anti-6,4-PP (Clone 64M-2, Supplier Cosmo Bio, CAC-NM-DND-002): https://www.cosmobiousa.com/products/anti-6-4pps-mab-clone-64m-2<br>- Anti phospho Histone H2A.X (Ser139) (Clone JBW301, Supplier Millipore, 05-636): https://www.sigmaaldrich.com/DE/en/product/sigma/zrb05636?gclid=CjwKCAiApvebBhAvEiwAe7mHSK2Re-XHNNGZdzDlNOYV6VBEZ6fk6ECEIn2eEcMLrH0usRhY6n4WnhoC7aMQAvD_BwE&gclsrc=aw.ds<br>- AffiniPure Peroxidase-conjugated secondary antibody (Polyclonal, Jackson Immuno Research, 115-035-174): https://www.jacksonimmuno.com/catalog/products/115-035-174<br>- Anti-mouse IgG Alexa Fluor 555 (Polyclonal, Thermo Fisher Scientific, A-21422): https://www.thermofisher.com/antibody/product/Goat-anti-Mouse-IgG-H-L-Cross-Adsorbed-Secondary-Antibody-Polyclonal/A-21422 |

## Eukaryotic cell lines

Policy information about cell lines and Sex and Gender in Research

| Cell line source(s) | U2OS (cell line from tibia sarcoma of a female osteosarcoma patient) - ATCC Cat. No. HTB-96. |
|---|---|
| Authentication | Commercially available, visually and phenotypically as expected, no further authentication was performed. |
| Mycoplasma contamination | Regular mycoplasma testing showed no signs of contamination throughout the experiments. |

| Commonly misidentified lines<br>(See ICLAC register) | No commonly misidentified lines were used. |
|---|---|

# Animals and other research organisms

Policy information about studies involving animals; ARRIVE guidelines recommended for reporting animal research, and Sex and Gender in Research

| Laboratory animals | Species:<br>- Caenorhabditis elegans, strains are N2, MT8839, MT10430, MT15107, MT8879, MT11147, JJ1549, BJS634, RB951, RB1789, PFR40, MT8189, MT14390, RB1801, FX03886, RB864, FX04539, BJS21, BJS631, BJS629, BJS630, BJS772, BJS825, DW102, BJS890, BJS868, FX1524, BJS887, XF132, BJS722, hermaphrodites, ages ranging from day 1 to death (approx. 30 days maximum)<br>- Mus musculus, strain  FVB/nj:C57BL/6j, age: 15 days old (postnatal) |
|---|---|
| Wild animals | No wild animals were used in this study |
| Reporting on sex | C. elegans were hermaphrodite for all experiments.<br>Mice:<br>For TUNEL assay, 3 WT (2M, 1F), 5 WT+H (3M, 2F), 7 ERCC1 KO (4M, 3F) and 7 ERCC1 KO+H (4M, 3F)<br>For γH2AX staining, 5 WT (3M, 2F), 5 WT+H (3M, 2F), 6 ERCC1 KO (3M, 3F) and 5 ERCC1 KO+H (3M, 2F) |
| Field-collected samples | No field-collected samples were used in this study |
| Ethics oversight | The animal facility at the Institute of Molecular Biology and Biotechnology (IMBB) operates in compliance with the "Animal Welfare Act" of the Greek government, using the "Guide for the Care and Use of Laboratory Animals" as its standard. Animal license 6ΛΤΑ7ΛΚ-ΚΚΘ issued by the Veterinary Medicine Directorate of Greek Republic. |

Note that full information on the approval of the study protocol must also be provided in the manuscript.

# Flow Cytometry

## Plots

Confirm that:

☒ The axis labels state the marker and fluorochrome used (e.g. CD4-FITC).

☒ The axis scales are clearly visible. Include numbers along axes only for bottom left plot of group (a 'group' is an analysis of identical markers).

☒ All plots are contour plots with outliers or pseudocolor plots.

☒ A numerical value for number of cells or percentage (with statistics) is provided.

## Methodology

| Sample preparation | U2OS were cultured in DMEM, high glucose GlutaMAX Supplement, Pyruvate (Thermo Fisher Scientific, 31966047) with 10% fetal bovine serum (FBS; Biochrom GmbH, S0615) and 1% Penicillin-Streptomycin (Thermo Fisher Scientific, 15140112). Cells were kept at 37 °C in a 5% $CO_2$ incubator (Binder). Cell dissociation from the plates was performed with Accutase (Sigma, A6964). To promote quiescence, cells were cultivated in FBS-free medium for 48 hours before genotoxic treatment were applied. 24 hours after FBS-free medium, cells were mock treated or received harmine hydrochloride (diluted in water) or INDY (diluted in DMSO) (Sigma, SMB00461 and SML1011) at 10 or 25 µM respectively. Before the genotoxic treatment, cells were washed with FBS-free medium. For the UV treatment, medium was removed from the plates and cells were irradiated using 254 nm UV-C light Phillips UV6 bulbs. The MMS treatment was performed by adding MMS at 2 mM for 2 hours, followed by 3 washes with FBS-free medium. Quantification via flow cytometry of cell death and apoptosis was performed 24 hours after genotoxic treatment. Collected cells were incubated in Annexin V Binding buffer (BioLegend, 422201) with Pacific Blue Annexin V (BioLegend, 640917) and 7-AAD (Thermo Fisher Scientific, 00699350) at 4 °C for 15 minutes. C |
|---|---|
| Instrument | Cells were measured using a MACSQuant VYB (Miltenyi Biotec). |
| Software | Cell data was obtained using MACSQuantify software 2.13.0, and analyzed using FlowJo (BD) v10.7.1. |
| Cell population abundance | Flow cytometry analysis was performed without sorting. |
| Gating strategy | Flow cytometry analysis started by selecting singlet events utilizing FSC-H vs FSC-A and selecting cells with a clear linear relation between height and area. FSC-A and SSC-A was utilized to separate debris, with low size and complexity, from the population of cells, which appeared clear. Finally, single cells were analyzed using the V1 channel (suitable for Pacific Blue, which is conjugated to the Annexin V) and the B2 channel (suitable for 7-AAD), as shown in the manuscript. The populations negative and positive for these markers were clearly defined, changed as expected upon DNA-damage-induced cell death and resembled similar experiments from the literature. |

☒ Tick this box to confirm that a figure exemplifying the gating strategy is provided in the Supplementary Information.

