## [Peer Review File · Nature Structural & Molecular Biology]

Peer Review Information

Manuscript Title: The DREAM complex functions as conserved master regulator of somatic DNA repair capacities

Corresponding author name(s): Björn Schumacher

Editorial Notes:

Redactions – transferred manuscripts (mention of previous referee reports from elsewhere)	This manuscript has been previously reviewed at another journal. This document only contains reviewer comments, rebuttal and decision letters for versions considered at Nature Structural and Molecular Biology. Mentions of prior referee reports have been redacted
--	--

Reviewer Comments & Decisions:

Decision Letter, initial version:
--

Message: Our ref: NSMB-A47218-T

10th Jan 2023

Dear Dr. Schumacher,

Thank you for submitting your revised manuscript "The DREAM complex functions as conserved master regulator of somatic DNA repair capacities" (NSMB-A47218-T). As conveyed when your manuscript transferred to us, given the positive reviews of referees 1 and 2, we are happy to accept it in principle in Nature Structural & Molecular Biology, pending minor revisions to satisfy the referees' final requests and to comply with our editorial and formatting guidelines.

Before we begin our detailed checks on your paper, you will have to fill in and upload the editorial policy checklist (available here: <https://www.nature.com/documents/nr-editorial-policy-checklist-flat.pdf>) to your manuscript via the online system or send it to us via email (nsmb@us.nature.com). Upon doing so, we will initiate our checks and then will send you a checklist with our editorial and formatting requirements in about a week. With the exception of the requested filled-in editorial policy checklist, please do not upload the final materials and make any revisions until you receive this additional information from us.

Thank you again for your interest in Nature Structural & Molecular Biology. Please do not hesitate to contact me if you have any questions.

Sincerely,

Dimitris Typas
Associate Editor
Nature Structural & Molecular Biology
ORCID: 0000-0002-8737-1319

Final Decision Letter:

Message 8th Feb 2023

:

Dear Prof. Schumacher,

We are now happy to accept your revised paper "The DREAM complex functions as conserved master regulator of somatic DNA repair capacities" for publication as a Article in Nature Structural & Molecular Biology.

To assist our authors in disseminating their research to the broader community, our SharedIt initiative provides all co-authors with the ability to generate a unique shareable link that will allow anyone (with or without a subscription) to read the published article.

Recipients of the link with a subscription will also be able to download and print the PDF.

Your paper will be published online soon after we receive proof corrections and will appear in print in the next available issue. You can find out your date of online publication by contacting the production team shortly after sending your proof corrections. Content is published online weekly on Mondays and Thursdays, and the embargo is set at 16:00 London time (GMT)/11:00 am US Eastern time (EST) on the day of publication. Now is the time to inform your Public Relations or Press Office about your paper, as they might be interested in promoting its publication. This will allow them time to prepare an accurate and satisfactory press release. Include your manuscript tracking number (NSMB-A47218A) and our journal name, which they will need when they contact our press office.

About one week before your paper is published online, we shall be distributing a press release to news organizations worldwide, which may very well include details of your work. We are happy for your institution or funding agency to prepare its own press release, but it must mention the embargo date and Nature Structural & Molecular Biology. If you or your Press Office have any enquiries in the meantime, please contact press@nature.com.

Please note that *Nature Structural & Molecular Biology* is a Transformative Journal (TJ). Authors may publish their research with us through the traditional subscription access route or make their paper immediately open access through payment of an article-

processing charge (APC). Authors will not be required to make a final decision about access to their article until it has been accepted. [Find out more about Transformative Journals](https://www.springernature.com/gp/open-research/transformative-journals)

Authors may need to take specific actions to achieve [compliance with funder and institutional open access mandates](https://www.springernature.com/gp/open-research/funding/policy-compliance-faqs). If your research is supported by a funder that requires immediate open access (e.g. according to [Plan S principles](https://www.springernature.com/gp/open-research/plan-s-compliance)) then you should select the gold OA route, and we will direct you to the compliant route where possible. For authors selecting the subscription publication route, the journal's standard licensing terms will need to be accepted, including [self-archiving policies](https://www.springernature.com/gp/open-research/policies/journal-policies). Those licensing terms will supersede any other terms that the author or any third party may assert apply to any version of the manuscript.

Sincerely,

Dimitris Typas
Associate Editor
Nature Structural & Molecular Biology
ORCID: 0000-0002-8737-1319